# OIDA-QA: A Multimodal Benchmark for Analyzing the Opioid Industry Document Archive

## Abstract

The opioid crisis is a serious public health issue that requires innovative solutions for effective analysis and deeper understanding. Despite the vast amounts of data in the Opioid Industry Documents Archive (OIDA), the complexity, multimodal nature, and specialized characteristics of healthcare data necessitate more advanced methods and models tailored to specific data types and detailed annotations, ensuring the precision and professionalism in the analysis. In this paper, we tackle this challenge by organizing the original dataset according to document attributes and constructing a benchmark with 400k training documents and 10k for testing. We extract extensive multimodal information from each document, including textual, visual, and layout information, to capture a wide range of features. Given the extracted dense information, we collect a comprehensive dataset comprising over 3 million question-answer pairs with the assistance of multiple AI models. We further develop domain-specific Large Language Models (LLMs) and investigate the impact of multimodal data on task performance. Our benchmarking and model efforts strive to produce an AI assistant system which can efficiently process the dataset and extract valuable insights. Preliminary results indicate the improvements with our AI assistant in document information extraction and question-answering tasks, highlighting the effectiveness of proposed benchmark in addressing the opioid crisis. The data and model will be made publicly available for research.

## 1 Introduction

The opioid crisis has significantly impacted global public health and revealed weaknesses in healthcare systems, as well as contributing to social and economic issues like increased domestic violence and child abuse (NIDA, 2024; Oderda et al., 2015; Swedo et al., 2020). In 2019, around 10.1 million Americans reported opioid misuse. Between June 2021 and May 2022, an estimated 108,000 drug overdose deaths occurred in the U.S., with opioids involved in 90% of these fatalities (CDC, 2020; Tanz et al., 2022). While effective for pain management, opioids can induce euphoria, resulting in misuse and addiction, particularly in regions with inadequate healthcare services, where individuals seek more accessible medical alternatives, underscoring the urgent need to address drug misuse (Birnbaum et al., 2011; Behzadi et al., 2018).

With the rapid development of big data and deep learning, artificial intelligence (AI) has shown great potential in healthcare, with applications in disease prediction, diagnostics, and personalized treatment. More recently, AI assistants based on Large Language Models (LLMs) have emerged as powerful tools for extracting insights from unstructured medical data and assisting healthcare professionals in clinical question answering (QA) and efficient information retrieval (Lee et al., 2020; Singhal et al., 2023; Lu et al., 2022). While the breakthroughs brought by LLMs are encouraging, due to historical reasons, much of the healthcare data still exists in scanned PDFs or image formats due to historical reasons, posing challenges for LLMs in processing non-pure text data (Van Landeghem et al., 2023; Gu et al., 2021). Additionally, concerns about data privacy and the unique nature of the healthcare field have resulted in a lack of publicly available data and detailed annotations, which has led to the underexplored application of AI technologies in addressing opioid misuse and addiction.

Fortunately, the public availability of data from the Opioid Industry Documents Archive (OIDA, 2021) presents a valuable opportunity to develop AI-driven solutions to address the opioid crisis. The OIDA provides millions of PDF documents that reveal information about many participants in the

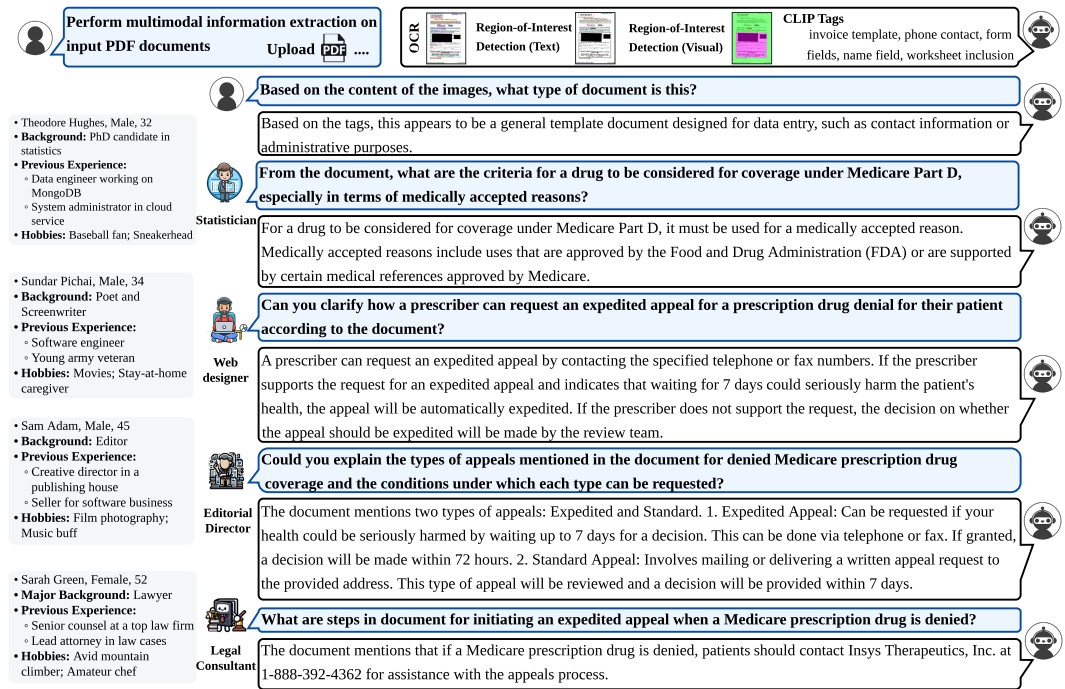

Figure 1: Overview of the proposed OIDA-QA: It is designed to handle the multimodal and multipage documents by extracting dense information (textual, visual, and layout) from scanned PDFs. The left side highlights our persona-based design, which enhances the system's capabilities, while the right side illustrates its effectiveness in addressing questions from a diverse range of users.

opioid industry during the U.S. opioid crisis. This resource enables us to study the crisis through documents released from opioid litigation and other sources (Alexander et al., 2022). However, the large scale and complexity of the OIDA present significant challenges. Most PDFs are scanned and contain multimodal elements such as text, figures, and tables spread across multiple pages, making it challenging to extract useful information for downstream tasks such as question answering (QA). Various methods have been proposed for this purpose, including Optical Character Recognition (OCR) (PaddleOCR, 2020; EasyOCR, 2020) and Object Detection(Pfitzmann et al., 2022; Zhong et al., 2019), among others. While effective for scientific papers, a more robust layout understanding model is required to interpret the visual elements in ODIA's data, including bar charts, tables, and analytical curves, *etc*.

Although LLMs excel at general QA problems, the multimodal long context of healthcare data presents challenges in model design, including long-context reasoning and the hallucinations associated with LLMs (Gao et al., 2023). Finally, the opioid crisis persists alongside the expanding OIDA data. Although the growing data volume is valuable, retraining the model with new data incurs significant time and financial costs. Thus, creating a low-cost, scalable model accessible to the public is essential.

In this paper, we propose OIDA-QA, a multimodel multi-page document question-answering benchmark based on the OIDA. To effectively handle large-scale document data, we start by analyzing its distribution using the taxonomy proposed in ADOPD (Gu et al., 2024), combined with the CLIP (Radford et al., 2021) model finetuned on ADOPD's image-caption pairs. Utilizing the taxonomy-derived clusters, we identify the 20 clusters with the most documents and further diversify based on subcategories and page count. We compile 20K PDF documents from each selected cluster to create the final training set. To develop a comprehensive understanding of each PDF document, we enrich the original PDFs by extracting textual information (OCR words), visual elements (tags and masks), and layout details (bounding boxes). These enriched, model-assisted multimodal annotations provide the foundation for generating QA pairs.

Unlike standard QA benchmarks that tend to ignore user attributes, this paper takes a different approach by considering the variety of questions users might ask, introducing persona-based question

generation. As shown in Figure 1, when generating questions for each PDF, we infer potential users through LLMs based on CLIP tags, then use persona data (Chan et al., 2024) to simulate different backgrounds, experiences, and hobbies, ensuring diverse perspectives in the generated questions. Meanwhile, the answerability of the questions is ensured by utilizing the LLM, which acts as the judge for the generated QA pairs. The QA pairs from different clusters fine-tune LLMs with the Low-Rank Adaptation (LoRA) method (Hu et al., 2022) for specialization. This class-aware LoRA design also enables leveraging closed LoRA with input PDFs during inference to complete tasks.

The contributions of this paper are summarized as follows:

- We introduce OIDA-QA, a multimodal document QA benchmark based on the OIDA, along with an effective method for enriching PDF documents with textual, visual, and layout annotations. This provides a data foundation for the exploration of AI-driven solutions to the opioid crisis.
- We utilize the dense extracted data and LLMs to create a persona-based QA benchmark. The integration of personas allows OIDA-QA to benefit a broader range of users.
- We develop a scalable model system that minimizes retraining costs while maintaining relevance and accessibility. Additionally, we provide a thorough analysis of different modalities and models, highlighting the current limitations and further proposing future research directions.

## 2 RELATED WORKS

**Document Understanding** Pre-training large models that handle both textual and visual information has proven highly effective for document understanding tasks (Huang et al., 2022; Da et al., 2023; Li et al., 2021; Gu et al., 2021). Unlike traditional LLMs that process plain text, document understanding requires models to consider layout information (Tu et al., 2023; Yu et al., 2023; Luo et al., 2023; Cao et al., 2023; Gu et al., 2023). Recently, LLMs and Multimodal LLMs (MLLMs) (OpenAI, 2022; 2023; Yang et al., 2023) have demonstrated outstanding zero-shot performance across a wide range of Natural Language Processing (NLP) and Computer Vision (CV) tasks. Leveraging LLMs for zero-shot document understanding has also shown promising progress (Perot et al., 2023; Zhang et al., 2023; Ye et al., 2023; Shi et al., 2023). For example, LLaVAR (Zhang et al., 2023) extends LLaVA (Liu et al., 2023b;a) to the document domain by pre-training with OCR data, where the fine-tuning document instructions are generated by GPTs (Achiam et al., 2023). Additionally, Qwen-VL (Bai et al., 2023) leverages document-level pre-training and direct QA for fine-tuning. Although existing models have demonstrated promising results in document tasks, most training data originates from traditional datasets that lack sufficient document data. The OIDA data poses challenges for existing models due to its multi-page, multimodal characteristics. Some works have been proposed for multi-image text generation tasks, such as InternVL (Chen et al., 2023b; 2024) and Pixtral (2024). However, these models were originally designed for general vision-language domain, not for multi-page document understanding tasks.

**LLMs in Healthcare** The advent of LLMs (Ouyang et al., 2022; Achiam et al., 2023; Dubey et al., 2024) has renewed interest in the possibilities of AI for medical services, which has been a long-term "grand challenge" (Shortliffe, 1987; Schwartz, 1987; Bobrow, 1994). The general purpose LLMs have achieved dramatic improvements on the medical benchmarks in the past months, such as MedQA (USMLE) (Jin et al., 2021), MedMCQA (Pal et al., 2022), and PubMedQA (Jin et al., 2019). *E.g.*, GPT 3.5 (Liévin et al., 2022) reached an accuracy of 50.82% on the MedQA dataset with the zero-shot in-context learning, Flan-PaLM reached an accuracy of 60.3%, and GPT-4-base (Nori et al., 2023) achieved 86.1% with the five-shot in-context learning. In parallel to the developments and evaluation of general-purpose LLMs on medical data, developing the medically specialized LLMs has been an ongoing effort. (Stanford CRFM, 2022; Wu et al., 2023) use GPT (Radford & Narasimhan, 2018) and LLaMA (Touvron et al., 2023b) to pretrain medical LLMs on medical domain text data . With the recent trend of scaling up pretraining data size and model parameter size, multiple studies explored the benefit of scaling up on medical tasks. Clinical-Camel (Toma et al., 2023) adapted from the LLaMA-2-70B (Touvron et al., 2023c) model using QLoRA (Dettmers et al., 2023) training on medical data. MEDITRON-70B (Chen et al., 2023a) scales up full-parameter medical domain pretraining to 70B parameters. LlamaCare (Li et al., 2024) fine-tune Llama 2 for the medical domain, using MIMIC-III (Johnson et al., 2016) as the training data and GPT-4 for diverse instruction generation. The most recent study Xie et al. (2024) shows that the state-of-the-art LLM o1 exhibits significant improvements over both the prior general-purpose LLMs and medical LLMs on

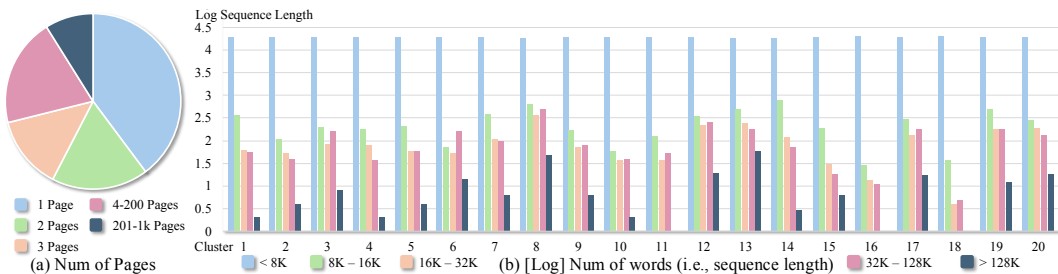

Figure 2: Data distribution of (a) number of pages and (b) number of words (i.e., sequence length).

medical question answering. However, some studies Anjum et al. (2024) find out the serious security issues of LLM-based healthcare due to hallucinations, which suggests that we should explore more trustworthy LLM techniques for medical applications.

## 3 OIDA-QA BENCHMARK

The original OIDA data is available in PDF format[1], along with metadata, low-resolution TIFF images, and extracted text for all documents. Despite the large volume of data, it does not provides layout and visual information due to the limitations of OCR and document extraction models. For the OIDA documents, Extracting as much detailed information as possible is essential to fully explore the model's capabilities and better serve the community. To fully leverage the OIDA for developing an AI system, our data collection process includes the following steps: (1) analyzing the distribution of the OIDA dataset, (2) performing balanced data sampling, and (3) extracting multimodal information.

### 3.1 DATA COLLECTION AND EXTRACTION

**Data Distribution Analysis.** To understand the distribution of OIDA, we utilize the pre-trained CLIP model from ADOPD and its taxonomy to tag the first page of each document in a zero-shot manner. This is achieved by calculating the similarity between the visual features of the page images and the textual embeddings. The textual embeddings are composed of the proposed taxonomy labels from ADOPD, formatted as '*a photo of <candidate label>*'. The top five most relevant labels are selected for document grouping, which categorizes each document into clusters based on the hierarchically structured ADOPD taxonomy. The predicted document tags and clustered results for all PDF documents offer a comprehensive overview of the OIDA dataset's distribution. This cluster-based sampling approach minimizes selection bias, resulting in a more diverse training set.

**Data Sampling.** Let the entire dataset be denoted by $\mathcal{D}_{\text{full}}$. We downsample $\mathcal{D}_{\text{full}}$ by selecting the top $K$ largest clusters based on distribution analysis above. For each cluster $k \in \{1, 2, \ldots, K\}$, we define the subset $\mathcal{D}_k = \{D_{k,1}, D_{k,2}, \ldots, D_{k,N_k}\}$, where each document $D_{k,i}$ consists of multiple pages. In our experiments, we set $K$ to 20. For the sampling of each cluster, we balance the subcategories according to the labels and the number of pages. We collect a diverse training set from these 20 clusters, including 20K PDF documents per cluster (for a total of 400K documents). For the test set, we gather 500 PDF documents per cluster, resulting in a total of 10K documents. In Figure 2(a), we visualize the distribution of documents across five different page counts within each cluster. Figure 2(b) visualizes the distribution of documents based on the logarithm of sequence length, emphasizing the long-sequence characteristics of OIDA-QA.

**Document Data Extraction.** For each document $D_{k,i}$, we extract textual, visual, and layout information from each page. Using an OCR tool, we extract words to obtain a set of text lines, which are then grouped into paragraphs $\{\mathbf{p}_{k,i,j}\}$ using heuristic rules. We assign each paragraph $\mathbf{p}_{k,i,j}$ a location $\mathbf{l}_{k,i,j} = \left(p_{k,i,j}, b_x^l, b_y^t, b_x^r, b_y^b\right)$, where $p_{k,i,j}$ is the page number, and $(b_x^l, b_y^t, b_x^r, b_y^b)$ are the normalized bounding box coordinates. Figure 3 illustrates that text lines alone do not capture the semantic relationships between words, and that merging words using rule-based methods is also limited by heuristic constraints. To address these limitations, we utilize the Doc2Box model (Gu et al., 2024) to extract text blocks that better preserve semantic structures. For visual information, we provide two outputs: CLIP tags and entity masks. The CLIP tags capture the high-level attributes of

---

[1] `s3://opioid-industry-documents-archive-dataset-bucket/`

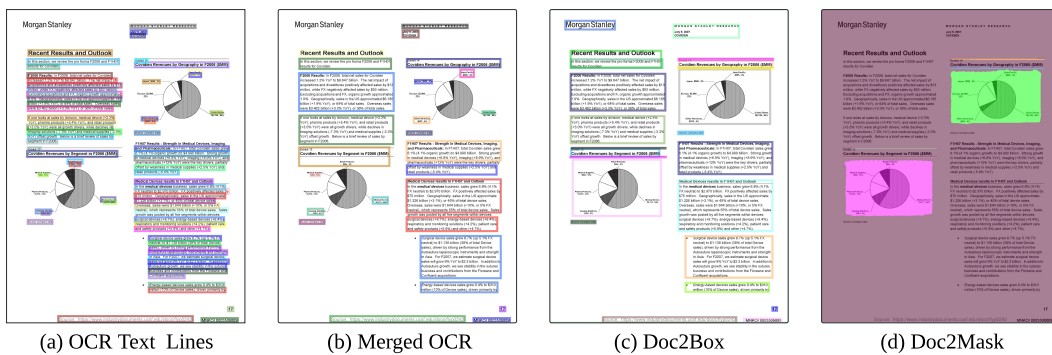

|(a) OCR Text Lines|(b) Merged OCR|(c) Doc2Box|(d) Doc2Mask|

Figure 3: Comparison of information extraction methods: (a) OCR text lines, (b) statistical merging results, (c) text blocks from Doc2Box, and (d) entity masks from Doc2Mask.

the documents. We apply the trained Doc2Mask model (Gu et al., 2024) to identify the entity masks. By combining textual, visual, and layout information, we create a comprehensive representation of each document that supports advanced processing tasks such as QA and information extraction.

## 3.2 PERSONA-BASED QUESTION-ANSWER DATASET CREATION

To simulate diverse user interactions and generate questions from various perspectives, we incorporate the vast Persona Hub (Chan et al., 2024), including over one billion personas, into our benchmark for question generation. We begin by generating the relevant personas for our benchmark. For each cluster, we randomly sample 500 personas from the full Persona Hub and employ the GPT-4o (OpenAI, 2023) to generate 48 detailed personas in average based on the assigned labels of the cluster with prompts detailed in Table 3 at Appendix. The detailed personas, including the attributes: *Name*, *Age*, *Gender*, *Major Background*, *Previous Experience*, and *Hobbies*, ensures the subsequent question generation process draws from a diverse and contextually relevant set of user profiles. More persona-based QA samples can be found in Figure 10 and Figure 11 in the Appendix.

**Algorithm 1:** QA Data Generation Process

**Input** : Document set $\mathcal{D}$;
    Persona pool $\mathcal{P}$;
    Desired number of QA pairs per document $N_{QA}$;
    Maximum attempts per document $M$
**Output** : QA dataset $\mathcal{QA}$
Initialize $\mathcal{QA} \leftarrow \emptyset$;
**foreach** *document* $D \in \mathcal{D}$ **do**
 Initialize QA count $n \leftarrow 0$, attempt count $m \leftarrow 0$;
 **while** $n < N_{QA}$ **and** $m < M$ **do**
  Sample personas $\mathcal{P}_D$ from $\mathcal{P}$ based on CLIP tags;
  **foreach** *persona* $P \in \mathcal{P}_D$ **do**
   Create prompt by combining document $D$ and persona $P$;
   Generate question $q$ and answer $a$ using GPT-4o;
   **if** *q is answerable* **then**
    Add $(q, a)$ to $\mathcal{QA}$;
    $n \leftarrow n + 1$;
  $m \leftarrow m + 1$;

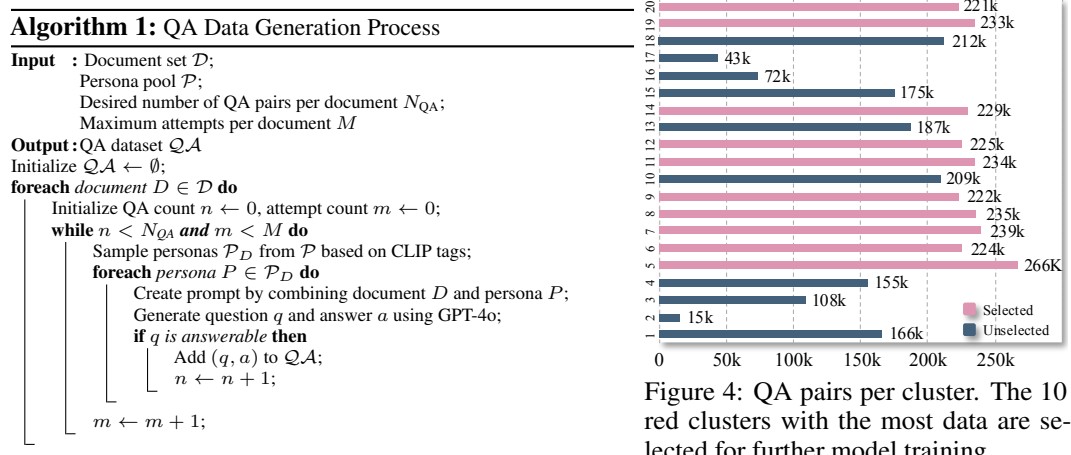

Figure 4: QA pairs per cluster. The 10 red clusters with the most data are selected for further model training.

Due to the large size and specialized nature of the OIDA dataset, hiring experts to create QA pairs is both costly and time-consuming. Instead, we employ a model-assisted approach for question-answer data generation, structuring the process into question generation and answer generation using GPT-4o. Algorithm 1 illustrates the QA data generation process. Building upon the data extraction mentioned in Section 3.1, we use only the grouped text lines as input for this step. To ensure a wide range of perspectives in the question generation process, we introduce the characterized personas discussed earlier. For generating QA pairs, we employ GPT-4o instances in the following role: (1) a question generator that creates questions based on the document content and persona attributes; (2) an answer generator that determines if the answer is answerable and provides the corresponding response simultaneously. The detailed prompts for question and answer generation are included in Table 4 and

Table 5 at Appendix, separately. As a result, over 3 million QA pairs are obtained and the number of QA pairs for each cluster is visualized in Figure 4.

# 4 METHOD

**Cluster-Specific LoRA Model Training** Let $f_\theta$ denote the pre-trained LLM with parameters $\theta$ (*e.g.*, LLaMA (Touvron et al., 2023a)). In the QA setting, a question $x$ is provided to the model to request an answer $y$, denoted as $y \sim f_\theta(y \mid x)$. Specifically, let $\mathbf{x}$ represents a sequence, and $x_i$ denotes its $i$-th token. $f_\theta$ generates text in an auto-regressive manner. Therefore, for a sequence $\mathbf{x} = \{x_1, x_2, \ldots, x_n\}$, the model $f_\theta$ computes the probability: $f_\theta(x) = \prod_{i=1}^{n} f_\theta(x_i \mid x_{1:i-1})$. The loss function for QA-based instruction tuning can be defined as:

$$L = -\frac{1}{n} \sum_{i=1}^{n} \log f_\theta(y_i \mid x, y_{1:i-1}), \tag{1}$$

where $n$ is the length of the response $y$.

To specialize the base model $f_\theta(\cdot)$ to each cluster $k \in \{1, 2, \ldots, K\}$, we train a separate LoRA adapter $\Delta\theta_k$ for each cluster while keeping the base parameters $\theta$ fixed. The adapted model for cluster $k$ becomes $f_{\theta+\Delta\theta_k}$. Given the QA dataset for cluster $k$, denoted as $\mathcal{QA}_k = \{(x_{k,i}, y_{k,i})\}$ with $N_k$ QA pairs, we train the adapter $\Delta\theta_k$ by minimizing the loss function specific to cluster $k$:

$$L_k = -\frac{1}{N_k} \sum_{(x,y) \in \mathcal{QA}_k} \sum_{i=1}^{n} \log f_{\theta+\Delta\theta_k}(y_i \mid x, y_{1:i-1}). \tag{2}$$

The training objective is to find the optimal adapter for each cluster, $\Delta\theta_k^* = \arg\min_{\Delta\theta_k} L_k$, by minimizing the loss $L_k$. By learning separate LoRA adapters for each cluster, we effectively capture knowledge from the data. As the dataset evolves, we can continuously add new clusters and train additional LoRA adapters. This approach enables each adapter to specialize in the unique characteristics of its cluster, while the shared base model $\theta$ retains general knowledge. With limited computational resources, our method offers a lower expected cost than fine-tuning the entire model.

**Inference with Cluster-Specific LoRA Models** During inference, we utilize the cluster-specific LoRA adapters to generate appropriate answers for a given input document $D_{\text{test}}$ and questions. We first extract key features from $D_{\text{test}}$, including OCR text, layout information, CLIP tags, and other visual captures. To find the most closest cluster for the test document $D_{\text{test}}$, we represent both $D_{\text{test}}$ and each cluster $k$ by averaging the embeddings of their labels. For $D_{\text{test}}$, we compute its vector representation as $\mathbf{v}_{\text{test}} = \sum_{t \in \mathcal{T}_{\text{test}}} \mathbf{e}(t)/|\mathcal{T}_{\text{test}}|$, where $\mathcal{T}_{\text{test}} = \{t_l\}_{l=1}^{L}$ denotes the set of CLIP tags in the test document and $\mathbf{e}(t)$ denotes the embedding of tag $t$. For each cluster $k$, the centroid vector is calculated as $\mathbf{c}_k = \sum_{t \in \mathcal{T}_k} \mathbf{e}(t)/|\mathcal{T}_k|$, with $\mathcal{T}_k$ being the set of all tags in cluster $k$. Consistent embeddings $\mathbf{e}(t)$ are crucial for meaningful similarity computations. These embeddings can be obtained using a pre-trained language model, such as Sentence-BERT (Reimers & Gurevych, 2019). We then compute the cosine similarity between $\mathbf{v}_{\text{test}}$ and $\mathbf{c}_k$ using $\text{sim}(D_{\text{test}}, k) = \frac{\mathbf{v}_{\text{test}} \cdot \mathbf{c}_k}{\|\mathbf{v}_{\text{test}}\| \|\mathbf{c}_k\|}$. The cluster with the highest similarity is selected: $k^* = \arg\max_k \text{sim}(D_{\text{test}}, k)$. Using the LoRA-adapted model for the selected cluster $k^*$, denoted as $f_{\theta+\Delta\theta_{k^*}}$, we generate the response $y_{\text{test}}$ conditioned on the input $x_{\text{test}}$, which includes extract document infomation: $y_{\text{test}} \sim f_{\theta+\Delta\theta_{k^*}}(y \mid x_{\text{test}})$. By leveraging the cluster-specific LoRA adapter $\Delta\theta_{k^*}$, the model incorporates specialized knowledge, thereby enhancing the relevance and accuracy of the generated response.

# 5 EXPERIMENTS

## 5.1 IMPLEMENTATION DETAILS

**Data and Training** To validate and demonstrate the effectiveness of our proposed dataset, we fine-tune Mistral-7B-Instruct (Jiang et al., 2023) model with torchtune (Meta, 2024) using LoRA. To streamline the fine-tuning process, we select the 10 clusters with the largest number of QA pairs and randomly sample 100k pairs for fine-tuning. The training is performed on NVIDIA A100 GPUs with

Table 1: Performance of Vicuna-7B with different input data configurations for 10 clusters through LoRA fine-tuning. "T" denotes text, "L" denotes layout, "P" denotes page, and "A" stands for tags.

| Model Vicuna-7B | T+L+P+A | | | | T+L+A | | | | T+L | | | | T+A | | | | T | | | |
|---|---|---|---|---|---|---|---|---|---|---|---|---|---|---|---|---|---|---|---|---|
| | BLEU | METEOR | ROUGE-L | BERTScore | BLEU | METEOR | ROUGE-L | BERTScore | BLEU | METEOR | ROUGE-L | BERTScore | BLEU | METEOR | ROUGE-L | BERTScore | BLEU | METEOR | ROUGE-L | BERTScore |
| $\mathcal{C}_{LoRA}^{5}$ | 33.91 | 55.67 | 51.19 | 78.73 | 34.05 | 55.80 | 51.31 | 78.76 | 33.86 | 55.82 | 51.32 | 78.72 | 33.84 | 55.21 | 50.18 | 78.21 | 33.56 | 55.81 | 51.06 | 78.57 |
| $\mathcal{C}_{LoRA}^{6}$ | 24.32 | 46.45 | 43.17 | 74.70 | 26.84 | 53.34 | 40.58 | 78.05 | 24.60 | 46.89 | 43.53 | 74.91 | 25.30 | 47.33 | 44.00 | 75.40 | 24.93 | 47.41 | 43.98 | 75.56 |
| $\mathcal{C}_{LoRA}^{7}$ | 23.73 | 46.47 | 41.78 | 75.27 | 18.79 | 41.88 | 38.55 | 74.70 | 23.82 | 46.71 | 41.89 | 75.37 | 23.95 | 46.78 | 41.86 | 75.51 | 23.95 | 46.78 | 42.00 | 75.59 |
| $\mathcal{C}_{LoRA}^{8}$ | 25.88 | 47.08 | 42.36 | 73.47 | 24.57 | 44.88 | 40.79 | 74.10 | 26.21 | 47.66 | 42.85 | 73.85 | 26.97 | 49.69 | 44.77 | 75.62 | 27.03 | 49.84 | 44.71 | 75.62 |
| $\mathcal{C}_{LoRA}^{9}$ | 24.93 | 46.87 | 42.35 | 74.20 | 25.05 | 46.99 | 42.67 | 74.37 | 25.03 | 47.09 | 42.62 | 74.39 | 25.37 | 48.09 | 43.49 | 75.14 | 25.33 | 48.03 | 43.35 | 75.04 |
| $\mathcal{C}_{LoRA}^{11}$ | 28.94 | 51.92 | 46.92 | 77.69 | 28.88 | 51.92 | 46.89 | 77.71 | 29.06 | 52.00 | 46.93 | 77.71 | 28.95 | 51.98 | 46.76 | 77.63 | 28.95 | 51.95 | 46.67 | 77.57 |
| $\mathcal{C}_{LoRA}^{12}$ | 21.63 | 43.76 | 40.32 | 71.40 | 22.09 | 44.27 | 40.75 | 71.73 | 21.74 | 44.09 | 40.50 | 71.60 | 22.44 | 46.00 | 42.87 | 73.47 | 22.68 | 46.19 | 42.63 | 73.53 |
| $\mathcal{C}_{LoRA}^{14}$ | 24.00 | 46.95 | 42.10 | 74.37 | 24.51 | 47.35 | 42.56 | 74.57 | 24.24 | 47.34 | 42.40 | 74.59 | 24.66 | 48.62 | 43.29 | 75.71 | 24.70 | 48.62 | 43.26 | 75.64 |
| $\mathcal{C}_{LoRA}^{19}$ | 20.86 | 42.72 | 38.48 | 70.90 | 21.07 | 42.96 | 38.71 | 71.20 | 19.59 | 42.42 | 38.29 | 71.37 | 26.03 | 48.12 | 43.95 | 75.53 | 26.12 | 48.54 | 44.05 | 75.61 |
| $\mathcal{C}_{LoRA}^{20}$ | 23.28 | 44.94 | 42.33 | 73.45 | 23.27 | 44.83 | 42.30 | 73.45 | 26.22 | 46.74 | 46.71 | 74.95 | 24.38 | 46.11 | 43.61 | 74.65 | 24.35 | 46.20 | 43.44 | 74.63 |
| Average | 25.15 | 47.28 | 43.10 | 74.42 | 24.91 | 47.42 | 42.51 | 74.86 | 25.44 | 47.68 | 43.70 | 74.75 | 26.19 | 48.79 | 44.48 | 75.69 | 26.16 | 48.94 | 44.52 | 75.74 |

80GB memory. For each model, we set the sequence length to 8,192 and use a batch size of 1. We integrate the LoRA modules into both the query and value matrices of the self-attention layers, as well as across all layers of the MLP modules. The rank parameter is configured to 8, and the scaling factor $\alpha$ is set to 16 to ensure optimal task performance. The models are optimized using the AdamW optimizer (Loshchilov & Hutter, 2018) with fused optimization, a learning rate of $3 \times 10^{-4}$, and a weight decay of 0.01. A cosine learning rate scheduler with 100 warmup steps is employed to ensure smooth transitions during training. We train the models for 5 epochs with gradient accumulation over 32 steps, utilizing a memory-efficient variant of the cross-entropy loss. This approach effectively handles large sequences despite the small batch size.

**Evaluation Metrics**   To assess the performance the models trained on OIDA-QA, we evaluate the answers using sentence-level automatic evaluation metrics through the Hugging Face evaluation pipeline (HuggingFace, 2024). The automated metrics include BLEU (Papineni et al., 2002), ME-TEOR (Banerjee & Lavie, 2005), ROUGE (Lin, 2004), and BERTScore (Zhang et al., 2019), which provide quantitative measures of the quality of generated answers by comparing them to reference answers. These metrics capture key aspects crucial for QA tasks: BLEU measures 4-gram overlap to assess lexical similarity, while METEOR considers both exact matches and synonyms, focusing on precision and recall. ROUGE evaluates sequence overlap, with ROUGE-L emphasizing the longest common subsequence. BERTScore utilizes pre-trained contextual embeddings to assess semantic similarity beyond exact matches[2]. This automatic sentence-level evaluation allows us to effectively analyze our model's performance in generating accurate and relevant answers.

## 5.2 Experimental Analysis

In this section, we evaluate the effectiveness of the proposed data and the models' performance trained under different settings. We begin by analyzing the effectiveness of each modality across different clusters, followed by a detailed analysis of the performance of both LLMs and MLLMs. Additionally, we assess our cluster-specific LoRA design during inference in cross-test settings.

**Effectiveness of Modalities for Different Clusters**   Considering that different modalities may contribute uniquely to specific clusters, we utilize the Vicuna-7B model, pre-trained on 125K tokens, and fine-tune LoRA adapters for 10 clusters using various input combinations, including text, layout, page, and tags. For ablation purposes, we fine-tune all Vicuna-7B LoRA models for just 1 epoch with a batch size of 32. As shown in Table 1, we examine 5 input configurations with Vicuna-7B model on 10 cluster data, where "T" denotes text, "L" denotes layout, "P" denotes page, and "A" denotes tags, respectively. On average, the text-only model achieves the best performance on BERTScore, whereas incorporating layout information tends to decrease overall performance. This is likely because LLMs are primarily trained on pure text, making it challenging for the model to learn layout information solely during the fine-tuning stage. Learning such information requires more

---

[2]BERTScore uses DeBERTa (https://huggingface.co/microsoft/deberta-xlarge-mnli) embeddings.

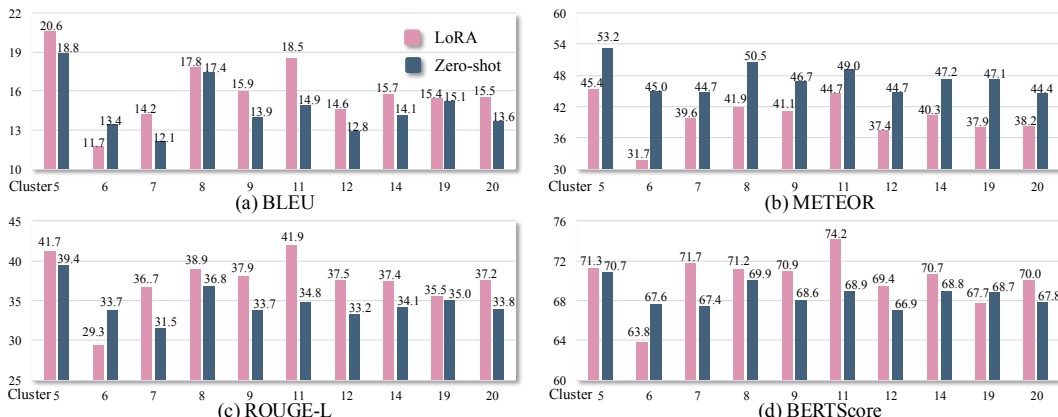

Figure 5: Evaluation results of 10 clusters with LoRA fine-tuned and zero-shot Mistral-7B model.

parameters for optimization and larger datasets. This challenge has also been noted in prior research on layout-aware LLMs (Gu et al., 2023; Fujitake, 2024).

We also find that for specific clusters, such as cluster 11, incorporating layout information improves the performance (BERTScore increased from 77.57 to 77.71). Cluster 11 encompasses various instructional and informational categories, including guides, tips, and tutorials. In these types of documents, layout information is crucial as they often contain substantial semi-structured data. Consequently, including layout information enhances the model's performance. In Table 1, cluster 5 achieves the best performance among all clusters. Cluster 5 consists mainly of design documents, which typically contain rich, detailed textual content and well-structured formats. Such documents provide ample contextual information and coherent narratives, making it easier for the language model to understand and generate accurate responses. The abundance of descriptive language and clear layout in design documents aligns well with the strengths of LLMs trained on large-scale text corpus. In contrast, cluster 19 exhibits the lowest performance, primarily comprising documents with addresses and contact information. These documents contain sparse, fragmented text and minimal contextual depth, making it challenging for the model to understand and respond accurately. The lack of descriptive language and coherent narratives contributes to the reduced performance in this cluster.

**Comparison of LLMs and MLLMs** We present the average evaluation results of 10 clusters across multiple metrics in Table 2, comparing Mistral-7B with and without LoRA, as well as zero-shot results and VLM performance using InternVL2-8B and Pixtral-12B. Detailed results for Mistral-7B across the 10 clusters are displayed in Figure 5. The LoRA fine-tuned models achieved high scores across the metrics compared to the zero-shot performance, indicating that models trained on our dataset are able to produce answers that are not only lexically similar to the reference answers but also semantically relevant and informative. Thus, the effectiveness of the models trained on our dataset is demonstrated on QA tasks. Specifically, higher BLEU score indicates better n-gram overlap and alignment with the reference answers; improvements in ROUGE-L and BERTScore show that the generated answers are more informative and semantically similar. We also observe that the zero-shot model appears to achieve the highest METEOR score, while the fine-tuned model seems to perform better on BERTScore. We think that METEOR may reward the semantic and lexical diversity present in the zero-shot outputs, whereas fine-tuning might reduce this diversity.

| Model | BLEU | METEOR | ROUGE-L | BERTScore |
|---|---|---|---|---|
| *LoRA-Trained* | | | | |
| Mistral-7B | 16.02 | 39.80 | 37.39 | 70.09 |
| *Zero-Shot* | | | | |
| Mistral-7B | 14.66 | 47.28 | 34.62 | 68.52 |
| *Vision-Language Models (Image Input Only)* | | | | |
| InternVL2-8B | 5.76 | 36.88 | 22.91 | 64.11 |
| Pixtral-12B | 7.23 | 36.02 | 25.90 | 61.89 |

Table 2: Comparison of the average performance with 10 clusters on the question-answering task: LoRA fine-tuned models on OIDA-QA vs. zero-shot LLMs and VLMs.

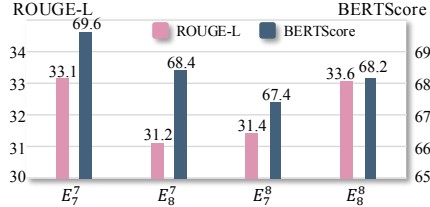

Figure 6: Ablation of the similarity design, $E_y^x$ denotes evaluation with cluster $x$ fine-tuned model on cluster $y$ data.

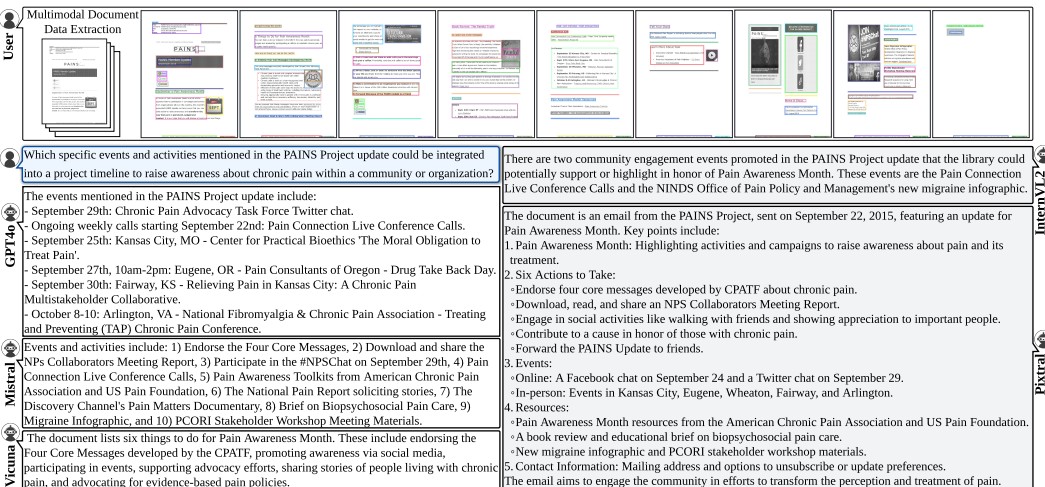

Figure 7: Visualization of responses from our fine-tuned model versus zero-shot MLLMs.

Interestingly, Table 2 shows that MLLMs, despite not being fine-tuned on OIDA-QA, achieve reasonable BERTScore performance. Pixtral-12B, though a newer model performing well on benchmarks like CountBenchQA (Deitke et al., 2024), performs worse on BERTScore for OIDA-QA compared to InternVL2. Pixtral utilizes a 400M parameter vision encoder, whereas InternVL2 uses a 300M parameter encoder distilled from a 6B parameter vision model extensively trained on OCR data, highlighting the importance of pre-training on document data. Additionally, Figure 7 shows that while Pixtral generates longer and more diverse responses, it also exhibits hallucination issues.

**Effectiveness of Cluster-Specific LoRA Design**   As introduced in Section 4, during inference, we select the most suitable cluster-specific LoRA adapter by matching the input document to the closest cluster centroid based on similarity. We further randomly select two different clusters (cluster 7 and cluster 8) and evaluate the performance of two corresponding LoRA fine-tuned model on the dataset with 1k samples collected from two different clusters, the results are shown in Figure 6. The task performance is superior when fine-tuned and evaluated on the same cluster compared to when fine-tuned and evaluated on different clusters. Thus, the experiments confirm that models employing the similarity design outperform those without it. The ability to adapt to domain-specific nuances enables the models to generate more accurate and contextually appropriate answers.

# 6 CONCLUSION AND FUTURE WORK

This paper introduces OIDA-QA, a multimodal QA benchmark designed specifically for OIDA document understanding. OIDA-QA comprises 400K training documents and 10K testing documents. In addition, we have collected over 3 million QA pairs generated by various models to enhance the dataset's diversity and robustness. Our experimental results demonstrate the effectiveness of both the benchmark and the AI assistant system in tackling QA tasks, presenting a promising approach to addressing the opioid crisis.

There are several promising future research directions. Integrating complex layout information into LLM pre-training is essential, as fine-tuning alone may not enable models to fully comprehend layout-aware language. Extending the context window is vital for handling documents with extensive text and images, allowing the model to process more details and achieve deeper comprehension. Developing specialized image encoders for high-resolution document images is crucial since current MLLMs struggle with these inputs. Implementing grounding tasks can enhance AI assistant user interaction by guiding users to key areas and linking responses directly to source documents. Lastly, model compression and acceleration techniques are important for optimizing deployment, enabling the AI assistant to run efficiently on PCs and mobile devices, thus increasing accessibility. While this paper introduces the benchmark, we believe it will inspire future studies, aligning with the goal of OIDA-QA: *to empower AI in helping people tackle the opioid crisis*.

REPRODUCIBILITY STATEMENT

To ensure the reproducibility of our research and the validity of the proposed dataset, we commit to making the dataset publicly available for research purposes. Upon publication, all components of the benchmark, including the raw data, annotations, and preprocessing scripts, will be released under a suitable open-access license. This will allow the research community to replicate our experiments, validate the results, and build upon our work.

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
