# A    APPENDIX

## A.1    PROMPTS

In this section, we outline the prompts used for persona, question, and answer generation. All data generation processes are conducted on the GPT-4o platform.

| | | Prompt Sentence |
|---|---|---|
| 👤 | 1 | You will be given a list of persona. The order of the item means nothing in the list. Your job is to integrate new $N$ personas from them. |
| 👤 | 2 | Each new persona retrieves one item from the given list as the major background and merges some of the rest items as the previous experience and hobby. You must NOT change anything of the items you select from the list. |
| 👤 | 3 | You will be provided with a special cluster of documents, where each document is classified under several tags. Your task is to summarize who might be most closely related to the description of the cluster, integrating personas that strongly align with the document tags. |
| 👤 | 4 | When doing this, consider the relevance of these personas to individuals who regularly engage with the items mentioned by these tags in their daily work. Ensure that the integrated personas relate to as many tags as possible from the list. |
| 👤 | 5 | A person can have multiple working experiences before the major career and they should be relevant. People's hobbies can be various. You also need to predict the name, gender, and age from the integrated persona. |
| 👤 | 6 | You should think step by step and try your best to be creative. All of the integrated personas should be very diverse but related to the corresponding profession. The profession can also be non-professional. |
| 👤 | 7 | The following is the list of personas: $\mathcal{P}$ |
| 👤 | 8 | The statistic of document tags you should focus on: $\mathcal{T}$ |
| 👤 | 9 | You should generate output in the following JSON format, for example: $\mathcal{E}$ |
| 👤 | 10 | According to the document tags, your $N$ personas are: |

Table 3: Prompt for persona generation.

As shown in Table 3, sampled persona list $\mathcal{P}$; document tags of cluster $\mathcal{T}$; and the number of persona to generate, $N$ are given. Besides, we also provide a list of examples $\mathcal{E}$ to help GPT-4o better understand the task and format of outputs.

| | | Prompt Sentence |
|---|---|---|
| 👤 | 1 | You are a PDF Reader AI Assistant. You will be given a long PDF document, a set of user professions, and background of the users. Assume you are the given users and ask a question for each user referring to your document content, specified profession and background. |
| 👤 | 2 | The users' professions and background are provided below: $\mathcal{P}$ |
| 👤 | 3 | Please generate question that meet the following criteria: 1. Personalized: The questions should align with the user's interests, status, and profession. Be sure do not explicitly mention about the personal interests, status, or profession in the question! 2. Comprehensive: The questions should cover useful information the profession may interested in according to his background. 3. Answerability: The questions should be answerable using only the information provided within the document, without requiring any external knowledge. 4. Diversity: Avoid to generate the similar questions with the same answers for different users. |
| 👤 | 4 | When generate questions, please avoid to generate the following questions as there are unanswerble: $\mathcal{Q}_{unans}$ |
| 👤 | 5 | The following is the document: $D + L$ |
| 👤 | 5* | The following is the document: $D$ |
| 👤 | 6* | Some potential attributes are also given as an support for you to better understand the doucments. The potential attributes of the document are: $\mathcal{T}$ |

Table 4: Prompt for question generation.

The prompt for question generation is shown in Table 4. Besides the sampled personas $\mathcal{P}$, variant resources can be used to guide GPT-4o to generate questions more accurately and relevantly. As shown in Table 4, document text $D$, document layout $L$. and document tag $\mathcal{T}$ are available for the question generate. To evaluate the performance of the prompt with different combinations of document resources, we conduct an evaluation with GPT-4 on the 3 data configurations – "ALL", "w/o tags", "w/o layout", which represent scenarios of prompt GPT-4 with text, tags and layout (1, 2, 3, 4, 5, 6* in Table 4); text and layout (1, 2, 3, 4, 5); and text and tags (1, 2, 3, 4, 5*, 6*) respectively. The evaluation measures question relevance to the document and personas, answer relevance to the document, and answer quality.

Shown in Figure 8, it's found that the configuration "w/o tags" achieves the best performance among the 3 test configurations. In the following process, we use this configuration (1, 2, 3, 4, 5) as our question generation prompt. In our answer generation, GPT-4o is required to predict the answerability together with the answer at the same time. Like the question generation, we combine the OCR text $D$ and layouts $L$ as the document resource to answer the question. If the question is not answerable, GPT-4o should explain why it can not be answered.

| | | Prompt Sentence |
|---|---|---|
| 👤 | 1 | You are a PDF Reader AI Assistant. You will be given a lengthy PDF document and a det of related question. Your job is to generate answers based on the content of the provided document only. |
| 👤 | 2 | Answer the Question: If you find relevant information in the document, provide a detail and accurate answer. Explain if Not Found: If you cannot find the answer, explain clearly why the information is not available or why the document does not contain the necessary details. Partial Answers: If the document contains partial information, provide what you can and explain what is missing. Your goal is to assist by either answering the question based on the document or by explaining why an answer could not be determined from the document. |
| 👤 | 3 | Here is the content: $D + L$ |
| 👤 | 4 | Here is the question set: $\mathcal{Q}$ |
| 👤 | 4 | Output the answers in a JSON format: "Question": the question you answer, "Answerability": if the question can be answered (YES/NO), "Answer": if YES, provide the answer, ELSE provide REASON |

Table 5: Prompt for answer generation.

Figure 8: GPT-4o-Based Evaluation for document input configurations. metrics "Que-Doc", "Que-Per", "Ans-Doc", and "Ans-Qual" represents question relevance to the document and personas, answer relevance to the document, and answer quality.

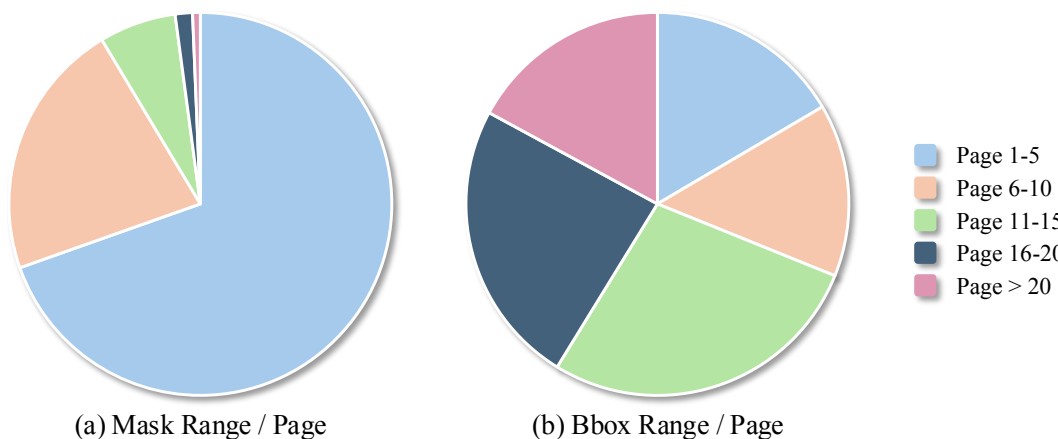

(a) Mask Range / Page          (b) Bbox Range / Page

Figure 9: The statistic of number of Text/Mask of OIDA. (a) shows the averaged number of Mask of per page. (b) shows the averaged number of Bbox of per page.

## A.2    DOCUMENT TEXT AND MASK

Figure 9 shows the statistic of the average number of text boundaries and masks on each page. Figure 9 We present the frequency rate of 5 ranges – 1-5, 6-10, 11-15, 16-20, and >20 masks per page for both 2 items. (a) presents the frequency distribution of the average number of masks per page across documents. It is found about 70% documents have 1-5 masks per page on average, and 22% documents contain an average of 6-10 masks. In Figure 9 (b), we present the distribution of the averaged number of text boundary boxes per page. The distribution of the average number of the boundary box is more balanced than that of the mask. As the most frequent category, the document with an averaged 11-15 boundary box contains approximately 28% of the dataset samples. While 16% of documents in the dataset contain 6-10 boundary boxes per page.

## A.3    QA EXAMPLES

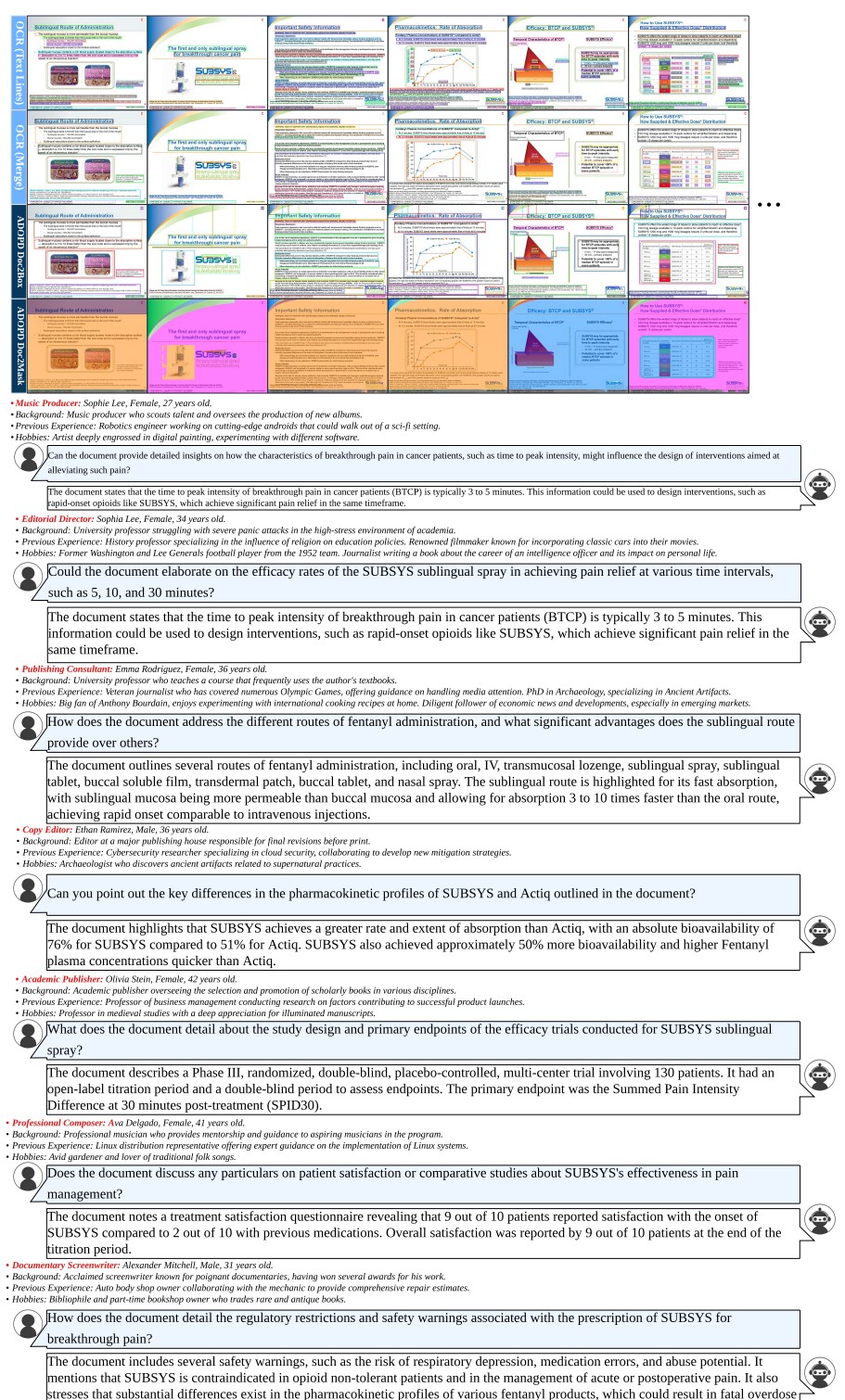

• **Music Producer:** *Sophie Lee, Female, 27 years old.*
• *Background: Music producer who scouts talent and oversees the production of new albums.*
• *Previous Experience: Robotics engineer working on cutting-edge androids that could walk out of a sci-fi setting.*
• *Hobbies: Artist deeply engrossed in digital painting, experimenting with different software.*

Can the document provide detailed insights on how the characteristics of breakthrough pain in cancer patients, such as time to peak intensity, might influence the design of interventions aimed at alleviating such pain?

The document states that the time to peak intensity of breakthrough pain in cancer patients (BTCP) is typically 3 to 5 minutes. This information could be used to design interventions, such as rapid-onset opioids like SUBSYS, which achieve significant pain relief in the same timeframe.

• **Editorial Director:** *Sophia Lee, Female, 34 years old.*
• *Background: University professor struggling with severe panic attacks in the high-stress environment of academia.*
• *Previous Experience: History professor specializing in the influence of religion on education policies. Renowned filmmaker known for incorporating classic cars into their movies.*
• *Hobbies: Former Washington and Lee Generals football player from the 1952 team. Journalist writing a book about the career of an intelligence officer and its impact on personal life.*

Could the document elaborate on the efficacy rates of the SUBSYS sublingual spray in achieving pain relief at various time intervals, such as 5, 10, and 30 minutes?

The document states that the time to peak intensity of breakthrough pain in cancer patients (BTCP) is typically 3 to 5 minutes. This information could be used to design interventions, such as rapid-onset opioids like SUBSYS, which achieve significant pain relief in the same timeframe.

• **Publishing Consultant:** *Emma Rodriguez, Female, 36 years old.*
• *Background: University professor who teaches a course that frequently uses the author's textbooks.*
• *Previous Experience: Veteran journalist who has covered numerous Olympic Games, offering guidance on handling media attention. PhD in Archaeology, specializing in Ancient Artifacts.*
• *Hobbies: Big fan of Anthony Bourdain, enjoys experimenting with international cooking recipes at home. Diligent follower of economic news and developments, especially in emerging markets.*

How does the document address the different routes of fentanyl administration, and what significant advantages does the sublingual route provide over others?

The document outlines several routes of fentanyl administration, including oral, IV, transmucosal lozenge, sublingual spray, sublingual tablet, buccal soluble film, transdermal patch, buccal tablet, and nasal spray. The sublingual route is highlighted for its fast absorption, with sublingual mucosa being more permeable than buccal mucosa and allowing for absorption 3 to 10 times faster than the oral route, achieving rapid onset comparable to intravenous injections.

• **Copy Editor:** *Ethan Ramirez, Male, 36 years old.*
• *Background: Editor at a major publishing house responsible for final revisions before print.*
• *Previous Experience: Cybersecurity researcher specializing in cloud security, collaborating to develop new mitigation strategies.*
• *Hobbies: Archaeologist who discovers ancient artifacts related to supernatural practices.*

Can you point out the key differences in the pharmacokinetic profiles of SUBSYS and Actiq outlined in the document?

The document highlights that SUBSYS achieves a greater rate and extent of absorption than Actiq, with an absolute bioavailability of 76% for SUBSYS compared to 51% for Actiq. SUBSYS also achieved approximately 50% more bioavailability and higher Fentanyl plasma concentrations quicker than Actiq.

• **Academic Publisher:** *Olivia Stein, Female, 42 years old.*
• *Background: Academic publisher overseeing the selection and promotion of scholarly books in various disciplines.*
• *Previous Experience: Professor of business management conducting research on factors contributing to successful product launches.*
• *Hobbies: Professor in medieval studies with a deep appreciation for illuminated manuscripts.*

What does the document detail about the study design and primary endpoints of the efficacy trials conducted for SUBSYS sublingual spray?

The document describes a Phase III, randomized, double-blind, placebo-controlled, multi-center trial involving 130 patients. It had an open-label titration period and a double-blind period to assess endpoints. The primary endpoint was the Summed Pain Intensity Difference at 30 minutes post-treatment (SPID30).

• **Professional Composer:** *Ava Delgado, Female, 41 years old.*
• *Background: Professional musician who provides mentorship and guidance to aspiring musicians in the program.*
• *Previous Experience: Linux distribution representative offering expert guidance on the implementation of Linux systems.*
• *Hobbies: Avid gardener and lover of traditional folk songs.*

Does the document discuss any particulars on patient satisfaction or comparative studies about SUBSYS's effectiveness in pain management?

The document notes a treatment satisfaction questionnaire revealing that 9 out of 10 patients reported satisfaction with the onset of SUBSYS compared to 2 out of 10 with previous medications. Overall satisfaction was reported by 9 out of 10 patients at the end of the titration period.

• **Documentary Screenwriter:** *Alexander Mitchell, Male, 31 years old.*
• *Background: Acclaimed screenwriter known for poignant documentaries, having won several awards for his work.*
• *Previous Experience: Auto body shop owner collaborating with the mechanic to provide comprehensive repair estimates.*
• *Hobbies: Bibliophile and part-time bookshop owner who trades rare and antique books.*

How does the document detail the regulatory restrictions and safety warnings associated with the prescription of SUBSYS for breakthrough pain?

The document includes several safety warnings, such as the risk of respiratory depression, medication errors, and abuse potential. It mentions that SUBSYS is contraindicated in opioid non-tolerant patients and in the management of acute or postoperative pain. It also stresses that substantial differences exist in the pharmacokinetic profiles of various fentanyl products, which could result in fatal overdose if not properly managed.

Figure 10: Visually-rich multipage document data extraction and persona-based QA data sample.

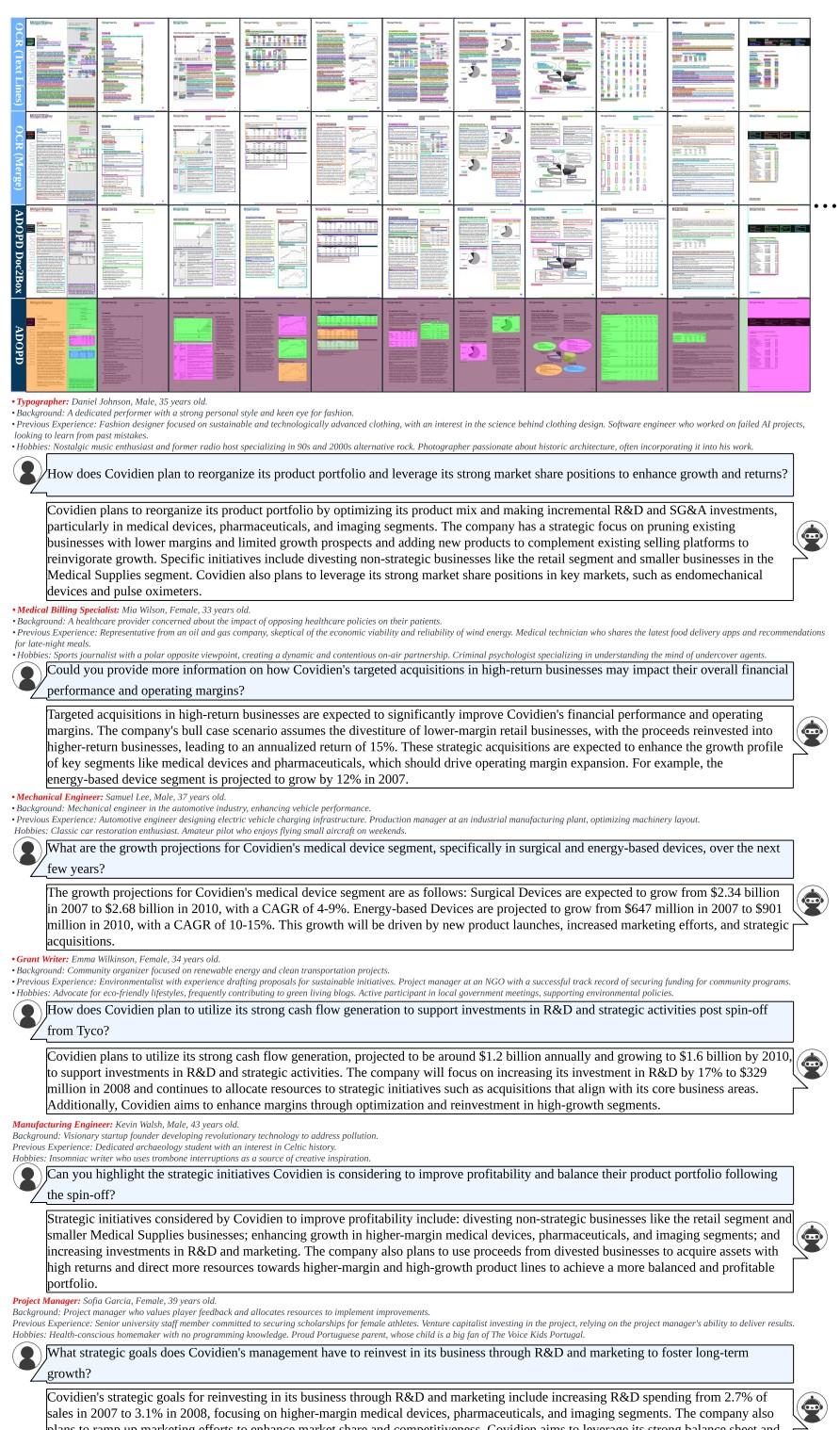

• **Typographer:** *Daniel Johnson, Male, 35 years old.*
• *Background: A dedicated performer with a strong personal style and keen eye for fashion.*
• *Previous Experience: Fashion designer focused on sustainable and technologically advanced clothing, with an interest in the science behind clothing design. Software engineer who worked on failed AI projects, looking to learn from past mistakes.*
• *Hobbies: Nostalgic music enthusiast and former radio host specializing in 90s and 2000s alternative rock. Photographer passionate about historic architecture, often incorporating it into his work.*

How does Covidien plan to reorganize its product portfolio and leverage its strong market share positions to enhance growth and returns?

Covidien plans to reorganize its product portfolio by optimizing its product mix and making incremental R&D and SG&A investments, particularly in medical devices, pharmaceuticals, and imaging segments. The company has a strategic focus on pruning existing businesses with lower margins and limited growth prospects and adding new products to complement existing selling platforms to reinvigorate growth. Specific initiatives include divesting non-strategic businesses like the retail segment and smaller businesses in the Medical Supplies segment. Covidien also plans to leverage its strong market share positions in key markets, such as endomechanical devices and pulse oximeters.

• **Medical Billing Specialist:** *Mia Wilson, Female, 33 years old.*
• *Background: A healthcare provider concerned about the impact of opposing healthcare policies on their patients.*
• *Previous Experience: Representative from an oil and gas company, skeptical of the economic viability and reliability of wind energy. Medical technician who shares the latest food delivery apps and recommendations for late-night meals.*
• *Hobbies: Sports journalist with a polar opposite viewpoint, creating a dynamic and contentious on-air partnership. Criminal psychologist specializing in understanding the mind of undercover agents.*

Could you provide more information on how Covidien's targeted acquisitions in high-return businesses may impact their overall financial performance and operating margins?

Targeted acquisitions in high-return businesses are expected to significantly improve Covidien's financial performance and operating margins. The company's bull case scenario assumes the divestiture of lower-margin retail businesses, with the proceeds reinvested into higher-return businesses, leading to an annualized return of 15%. These strategic acquisitions are expected to enhance the growth profile of key segments like medical devices and pharmaceuticals, which should drive operating margin expansion. For example, the energy-based device segment is projected to grow by 12% in 2007.

• **Mechanical Engineer:** *Samuel Lee, Male, 37 years old.*
• *Background: Mechanical engineer in the automotive industry, enhancing vehicle performance.*
• *Previous Experience: Automotive engineer designing electric vehicle charging infrastructure. Production manager at an industrial manufacturing plant, optimizing machinery layout.*
 *Hobbies: Classic car restoration enthusiast. Amateur pilot who enjoys flying small aircraft on weekends.*

What are the growth projections for Covidien's medical device segment, specifically in surgical and energy-based devices, over the next few years?

The growth projections for Covidien's medical device segment are as follows: Surgical Devices are expected to grow from $2.34 billion in 2007 to $2.68 billion in 2010, with a CAGR of 4-9%. Energy-based Devices are projected to grow from $647 million in 2007 to $901 million in 2010, with a CAGR of 10-15%. This growth will be driven by new product launches, increased marketing efforts, and strategic acquisitions.

• **Grant Writer:** *Emma Wilkinson, Female, 34 years old.*
• *Background: Community organizer focused on renewable energy and clean transportation projects.*
• *Previous Experience: Environmentalist with experience drafting proposals for sustainable initiatives. Project manager at an NGO with a successful track record of securing funding for community programs.*
• *Hobbies: Advocate for eco-friendly lifestyles, frequently contributing to green living blogs. Active participant in local government meetings, supporting environmental policies.*

How does Covidien plan to utilize its strong cash flow generation to support investments in R&D and strategic activities post spin-off from Tyco?

Covidien plans to utilize its strong cash flow generation, projected to be around $1.2 billion annually and growing to $1.6 billion by 2010, to support investments in R&D and strategic activities. The company will focus on increasing its investment in R&D by 17% to $329 million in 2008 and continues to allocate resources to strategic initiatives such as acquisitions that align with its core business areas. Additionally, Covidien aims to enhance margins through optimization and reinvestment in high-growth segments.

**Manufacturing Engineer:** *Kevin Walsh, Male, 43 years old.*
*Background: Visionary startup founder developing revolutionary technology to address pollution.*
*Previous Experience: Dedicated archaeology student with an interest in Celtic history.*
*Hobbies: Insomniac writer who uses trombone interruptions as a source of creative inspiration.*

Can you highlight the strategic initiatives Covidien is considering to improve profitability and balance their product portfolio following the spin-off?

Strategic initiatives considered by Covidien to improve profitability include: divesting non-strategic businesses like the retail segment and smaller Medical Supplies businesses; enhancing growth in higher-margin medical devices, pharmaceuticals, and imaging segments; and increasing investments in R&D and marketing. The company also plans to use proceeds from divested businesses to acquire assets with high returns and direct more resources towards higher-margin and high-growth product lines to achieve a more balanced and profitable portfolio.

**Project Manager:** *Sofia Garcia, Female, 39 years old.*
*Background: Project manager who values player feedback and allocates resources to implement improvements.*
*Previous Experience: Senior university staff member committed to securing scholarships for female athletes. Venture capitalist investing in the project, relying on the project manager's ability to deliver results.*
*Hobbies: Health-conscious homemaker with no programming knowledge. Proud Portuguese parent, whose child is a big fan of The Voice Kids Portugal.*

What strategic goals does Covidien's management have to reinvest in its business through R&D and marketing to foster long-term growth?

Covidien's strategic goals for reinvesting in its business through R&D and marketing include increasing R&D spending from 2.7% of sales in 2007 to 3.1% in 2008, focusing on higher-margin medical devices, pharmaceuticals, and imaging segments. The company also plans to ramp up marketing efforts to enhance market share and competitiveness. Covidien aims to leverage its strong balance sheet and cash flow generation to support these investments, thereby fostering long-term growth.

Figure 11: Text-rich multipage document data extraction and persona-based QA data sample.

| | Our Answer Wins | Tie | Ground Truth Wins |
|---|---|---|---|
| Cluster 18 | 25.7% | 32.5% | 41.8% |
| Cluster 19 | 17.9% | 26.1% | 56.0% |
| Cluster 20 | 20.1% | 36.2% | 43.7% |
| Cluster 29 | 16.9% | 32.5% | 50.6% |
| Cluster 34 | 25.6% | 27.5% | 46.9% |
| Cluster 38 | 25.4% | 31.5% | 43.1% |
| Cluster 39 | 18.6% | 20.9% | 60.5% |
| Cluster 65 | 22.2% | 29.4% | 48.4% |
| Cluster 90 | 21.2% | 25.9% | 52.9% |
| Cluster 94 | 21.1% | 27.9% | 51.0% |
| Average | 21.5% | 29.1% | 49.4% |

(a)

| | Persona Wins | Tie | No Persona Wins |
|---|---|---|---|
| Cluster 18 | 74.9% | 18.3% | 6.8% |
| Cluster 19 | 73.1% | 19.5% | 7.4% |
| Cluster 20 | 73.7% | 17.6% | 8.7% |
| Cluster 29 | 72.1% | 19.7% | 8.2% |
| Cluster 34 | 75.1% | 17.5% | 7.4% |
| Cluster 38 | 75.6% | 17.5% | 6.9% |
| Cluster 39 | 74.0% | 18.5% | 7.5% |
| Cluster 65 | 72.1% | 19.6% | 8.3% |
| Cluster 90 | 75.7% | 16.7% | 7.6% |
| Cluster 94 | 75.7% | 16.1% | 8.2% |
| Average | 74.2% | 18.1% | 7.7% |

(b)

Figure 12: (a) Utilize GPT-4 OpenAI (2023) to access answers generated by our model or the ground-truth responses from the test dataset. (b) Adopt GPT4o to access the questions generated with or without Personas Chan et al. (2024).

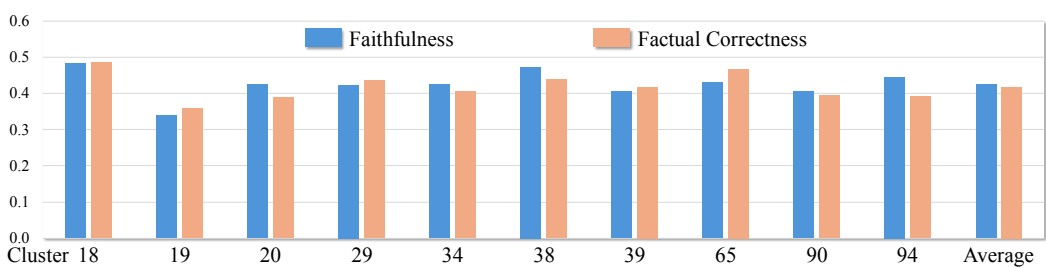

Figure 13: Results of faithfulness and factual correctness.

# B  ADDITIONAL EXPERIMENTS

## B.1  GPT-4 ASSESSMENT

**Quality of Generated Answers**  We utilize GPT-4 OpenAI (2023) to evaluate the quality of answers generated by our model in comparison to the ground-truth responses. As shown in Figure 12 (a), the overall average results indicate that our model's generated answers achieve a quality level comparable to the ground-truth, as assessed by GPT-4.

**Quality of Generated Questions**  We further employ GPT-4 to evaluate the quality of questions generated with and without personas. The results, illustrated in Figure 12 (b), reveal that questions generated with personas exhibit significantly higher quality compared to those generated without personas. This highlights the critical role of personas in question generation and underscores the effectiveness of our proposed approach.

## B.2  ADDITIONAL METRICS

We evaluate the faithfulness (F1 score) and factual correctness (using Ragas ragas (2024)) of our model's answers on the test dataset, as shown in Figure 13. Faithfulness, calculated at the token level, measures alignment with ground-truth responses, while factual correctness assesses adherence to verified facts. The results indicate that our model performs well on both metrics, further emphasizing its ability to generate good answers, especially when considering the BertScore results presented in Figure 5.