# OpenReview forum: "OIDA-QA: A Multimodal Benchmark for Analyzing the Opioid Industry Document Archive"
_ICLR.cc/2025/Conference — Submitted to ICLR 2025_

### Official Review · Reviewer_J2Vt · 2024-10-23

**Soundness:** 3
**Presentation:** 3
**Contribution:** 3
**Rating:** 5
**Confidence:** 3

**Summary:**

This paper proposes a new dataset, OIDA-QA, an Opioid Industry Document Archive Question-Answering dataset, which consists of over 400k samples with multimodal information (textual, visual, and layout) using an automated process of generating QA pairs using GPT-4o, while considering various types of questions for different human personas. For the experiments, the authors implement multiple models tailored to different document taxonomies and demonstrated that their proposed cluster-specific modeling approach can be a promising method for tackling QA for multimodal document understanding.

**Strengths:**

- The paper is well-motivated and easy to follow.
- This can be a significant contribution to multimodal document understanding.

**Weaknesses:**

- What is the procedure to ensure the quality of the QA data? I am sure it required tremendous effort just to create the data itself, but at least the test set should require some verification by humans or other solutions to ensure that a higher score truly reflects true positives and not false negatives.
- Although the authors show various performances, it would be helpful to have a reference for what is a 'good' score. For example, achieving some score could indicate that the model outperforms humans or some state-of-the-art models like GPT-4 in the main results table. This can further help readers to grasp what these score mean.
- More direct applications for solving this task in terms of machine learning (perhaps separated by each cluster). For instance, scoring high on questions in cluster 1 indicates that the model requires capabilities A, B, and C.
- How do we determine which category new questions belong to? Also, what is the rationale for disregarding unused clusters, rather than merging them into an 'out-of-distribution' cluster? My concern is not all questions may fit neatly into the 10 predefined clusters, especially if we aim to tackle the opioid crisis with AI-driven automation.

**Questions:**

- What are some representative examples from each cluster? Also, what insights can we gain from a model that achieves high performance on questions in, say, cluster 1?
- I am willing to raise the score if my concerns in the weakness and question parts are resolved.

---

> ### Author Response · Authors · 2024-11-25
> **Author Response to Q1 and Q2**
>
> We are grateful for the reviewer’s constructive critique and thoughtful input.
>
> ## Q1: Ensure the Quality
>
> Great question! Here, we highlight our efforts to ensure the quality of the QA data
>
> - *Long-Tail Coverage*: As shown in **Figure 2**, while most documents are fewer than 10 pages, long documents are crucial for building high-quality, long-context QA. We used a combination of manual annotation and automated sampling to enhance the representation of long-tail samples. This effort extends beyond document length to label types (e.g., emails, reports). For sensitive and low-frequency categories, such as prescriptions and non-typical opioid misuse patterns, we employed a human-in-the-loop approach to improve coverage.
>
> - Data Diversity: To adapt to various scenarios, the dataset incorporates diverse language styles, document types, and question-answer pairs. As detailed in **Section 3.1**, we curated data using taxonomy discovery, clustering, and human-in-the-loop processes to ensure inclusion from different sources (e.g., PDFs, handwritten notes, scanned tables). QA diversity was further enhanced with persona-based generation, providing varied perspectives and enriching question relevance.
>
> - Dataset Reliability: While we primarily relies on automated evaluation metrics in our test set, we ensured its reliability through well-designed annotation rules and filtering processes. For instance, we conducted an ablation study to assess the impact of personas on question generation (**Figure 12(b) or following table**), which showed significant improvements in diversity and relevance. However, due to resource constraints, we could not implement comprehensive human verification. We recognize the importance of this step and plan to include expert reviews or manual validation in future iterations to further enhance the quality of the test set.
>
> | Cluster | Persona Win | Tie   | No Persona Win |
> |---------|-------------|-------|----------------|
> | 18      | 74.9        | 18.3  | 6.8            |
> | 19      | 73.1        | 19.5  | 7.4            |
> | 20      | 73.7        | 17.6  | 8.7            |
> | 29      | 72.1        | 19.7  | 8.2            |
> | 34      | 75.1        | 17.5  | 7.4            |
> | 38      | 75.6        | 17.5  | 6.9            |
> | 39      | 74          | 18.5  | 7.5            |
> | 65      | 72.1        | 19.6  | 8.3            |
> | 90      | 75.7        | 16.7  | 7.6            |
> | 94      | 75.7        | 16.1  | 8.2            |
> | **average** | 74.2        | 18.1  | 7.7            |
>
>
> ## Q2: Winning rate
> Nice suggestion! To provide a clearer reference for what constitutes a "good" score, we utilized GPT-4 to evaluate the generated answers, with the results presented in **Figure 12(a)** in the supplementary material or **following table**. On average, across all clusters, our method achieves performance comparable to the ground-truth answers generated by GPT-4. Moreover, in most clusters, our model matches or even surpasses the quality of these ground-truth answers, underscoring its effectiveness and reliability. This comparison helps establish a meaningful benchmark for interpreting the scores.
>
> | Cluster | Our Answer Win | Tie   | Ground-Truth Win |
> |---------|----------------|-------|------------------|
> | 18      | 25.7           | 32.5  | 41.8             |
> | 19      | 17.9           | 26.1  | 56               |
> | 20      | 20.1           | 36.2  | 43.7             |
> | 29      | 16.9           | 32.5  | 50.6             |
> | 34      | 25.6           | 27.5  | 46.9             |
> | 38      | 25.4           | 31.5  | 43.1             |
> | 39      | 18.6           | 20.9  | 60.5             |
> | 65      | 22.2           | 29.4  | 48.4             |
> | 90      | 21.2           | 25.9  | 52.9             |
> | 94      | 21.1           | 27.9  | 51               |
> | **average** | 21.47          | 29.04 | 49.49            |

---

> ### Author Response · Authors · 2024-11-25
> **Author Response to Q3 and Q4**
>
> ## Q3: Cluster-specific capabilities
> We designed cluster-specific categorization based on our expertise in document processing and healthcare contexts. Different document types require distinct capabilities: text-rich documents (e.g., emails, logs) demand strong pure-text understanding, while semi-structured documents (e.g., tables, forms) require robust layout-aware NLP capabilities. For visually-rich documents, effective extraction of visual elements, facilitated by features like doc2box and doc2mask, becomes critical. By organizing data into clusters, we ensure that models develop the appropriate capabilities to process each document type effectively. Our ablation study, presented in **Table 1**, highlights how modalities like text, layout, page, and tags influence model performance. For instance, text-only models perform well on BERTScore, but integrating layout information can sometimes lower performance, likely due to LLMs being pre-trained primarily on pure text.
>
> However, for certain clusters, specific modalities enhance performance. For example, in **Cluster 11** (instructional documents like guides or tutorials), layout information proves beneficial due to their structured format. In **Cluster 9**, the inclusion of tag information extracted via the CLIP model boosts performance, as many documents in this cluster feature prominent visual elements. These findings underline the importance of tailoring modalities to the needs of each cluster. To facilitate further research, we will release the taxonomy used for clustering, along with document-specific tags, to support better exploration and model optimization.
>
> | Text | Layout | Tag | Page Information | Cluster 11 BERTScore |
> |------|--------|-----|------------------|-----------|
> | ✔    | ✔      | ✔   | ✔               | 77.69     |
> | ✔    | ✔      | ✔   | -               | 77.71     |
> | ✔    | ✔      | -   | -               | 77.71     |
> | ✔    | -      | ✔   | -               | 77.63     |
> | ✔    | -      | -   | -               | 77.57     |
>
> | Text | Layout | Tag | Page Information | Cluster 9 BERTScore |
> |------|--------|-----|------------------|-----------|
> | ✔    | ✔      | ✔   | ✔               | 74.20     |
> | ✔    | ✔      | ✔   | -               | 74.37     |
> | ✔    | ✔      | -   | -               | 74.39     |
> | ✔    | -      | ✔   | -               | 75.14     |
> | ✔    | -      | -   | -               | 75.04     |
>
>
> ## Q4: out-of-distribution
> That’s an excellent question. Handling Out-of-Distribution (OOD) data in document-based tasks presents unique challenges. As highlighted in *[1]*, feature-level methods such as KNN-based approaches are particularly effective for detecting OOD instances. In our work, we focus on in-distribution (ID) clusters to provide a robust QA foundation. For example, Cluster 5, achieving the highest performance as shown in **Table 1**, contains design documents with detailed textual content and structured formats. These characteristics align with the strengths of LLMs, enabling accurate responses. Although the current 10 predefined clusters are not exhaustive, they effectively cover general categories while offering scalability for future extensions.
>
> To address OOD data, our cluster-specific LoRA design, inspired by *[2]*, provides two key strategies. First, we can map OOD data to the nearest ID cluster and apply its corresponding LoRA model. Second, for OOD data significantly different from ID clusters, we can train a Mix-LoRA model by combining existing LoRA models with a small amount of new OOD data. Additionally, as the OIDA dataset is updated, we will expand OIDA-QA with new clusters and models, leveraging the modularity of LoRA to adapt efficiently without retraining the entire system. This ensures the framework remains scalable and responsive to real-world challenges like the opioid crisis.
>
> *[1] A Critical Analysis of Document Out-of-Distribution Detection*
>
> *[2] LoraHub: Efficient Cross-Task Generalization via Dynamic LoRA Composition*

---

### Official Review · Reviewer_YrYP · 2024-10-31

**Soundness:** 2
**Presentation:** 2
**Contribution:** 2
**Rating:** 3
**Confidence:** 4

**Summary:**

The paper aims to build a AI models to assist information extraction and question-answering on Opioid Industry Documents Archive (OIDA).  To address this the paper introduces a methodology to automatically generate question-answer pairs using AI models which are then leveraged to train AI assistants.  The method follows these steps (1) Extract multimodal  (text, visual, layout etc) information for each PDF document (2) Cluster the documents based on structured taxonomy (3) For every cluster, sample personas from a persona Hub and  generate QA pairs using AI models given the persona of the user asking the question (3) Utilize the QA pairs to train QA models using LLMs.

**Strengths:**

The paper explores an innovative domain by advancing question-answering capabilities on PDFs that contain multimodal information. It also introduces a novel methodology to effectively address this challenge.

**Weaknesses:**

Its unclear how useful the overall generation framework is and would be helpful to address the following questions

(1) Its unclear how and if the generated questions are closer to real user questions. Some form of human evaluation or user rating od the system would give a better idea.
(2) What is the effect of using these different modalities in the generation? Can you perform an ablation study, where you remove the information from one of the modalities (text or image etc) and how does this effect the generation process, downstream model performance and hence user satisfaction ?
(3) How generalizable is the framework ? What other related domains can this framework be applied to  and what are limitations?

**Questions:**

It would be helpful to receive a response from the authors on the questions described above.

---

> ### Author Response · Authors · 2024-11-25
> **Author Response to Q1**
>
> We deeply value the reviewer’s input and their careful assessment of our work.
>
> ## Q1: Generated questions vs Real questions
>
> Great question! Ensuring that the generated questions closely simulate real user inquiries is a key motivation behind our persona-based approach. During the data collection phase, we conducted a study and observed that different users tend to ask different types of questions. For instance, general users often pose broad questions such as:
>
> ```
> Question: Simplify this document for me
> Question: What are the main themes of the document?
> Question: Outline the main sections of the document
> ````
>
> From the questions above, it is evident that achieving diverse data annotation is challenging and requires a deep understanding of the documents. Additionally, individuals have varying educational backgrounds and subjective perspectives, which further complicates the task of generating questions that effectively capture diverse user intents. Due to the limitations of user diversity, we augmented question generation using personas and GPT. Our sample results indicate that GPT-generated questions cover most of the real user questions. Furthermore, these questions align closely with those posed by medical domain experts during our actual testing, demonstrating the effectiveness of our approach in bridging user intent and system-generated outputs. Here are some actual tests.
>
> Here are some actual QA questions collected for the given document [1]
>
>
> ```
> Questions generated by GPT:
> 1. Question: What was Alice Lum’s rationale for targeting Dr. Judson Somerville as a key prescriber for Exalgo?
> 2. Question: Why might doctors have been willing to work with Mallinckrodt despite potential legal and ethical risks?
> 3. Question: What implications arise from Alice Lum identifying Dr. Judson Somerville as a prescriber who believed “pain medications do not create addicts”?
> 4. Question: How did Larry McClure’s sales script training use Dr. Griffin’s example to motivate other sales representatives?
> 5. Question: What risks might patients face when prescribed medications by doctors under scrutiny for overprescribing opioids?
> 6. Question: How might Mallinckrodt have benefited from Dr. Hoffberg’s advocacy for the chronic use of long-acting opioid formulations?
> ```
>
>
>
> ```
> Questions generated by human:
> 1. Question: What is Art Morelli’s relationship with Alice Lum?
> 2. Question: Why might Dr. Eugene Gosy and the Gosy Center have been “under a bit of scrutiny”?
> 3. Question: Why might Dr. Eugene Gosy have been interested in working with Mallinckrodt’s C.A.R.E.S. Alliance?
> 4. Question: How did Medical Affairs collaborate with the sales department?
> 5. Question: Why might Dr. Eugene Gosy’s “issues with hydromorphone” been a sales problem regarding Exalgo?
> 6. Question: What kind of data did the Mallinckrodt/Covidien sales team use when preparing to meet with prescribers?
> 7. Question: What kind of concerns did sales representatives have about insurance?
> 8. Question: What vendor provided some of the data Mallinckrodt/Covidien used to inform sales work?
> ```
>
> The examples above demonstrate that GPT-generated questions closely align with human-generated questions.
>
> However, it is important to note that we recognize users, when generating the actual questions, do not rely solely on a single document. Instead, these questions (Questions 6,7,8 generated by human) often synthesize information across multiple documents, as the example document contains various links to additional sources. This highlights the interconnected nature of the information and the need for broader context when addressing such questions.
>
> Following your suggestion, we conducted an ablation study comparing question generation with and without personas, using GPT-4 to evaluate question quality. The results, presented in **Figure 12(b)** of the supplementary material or the **table below**, show that incorporating personas enhances the naturalness and relevance of the questions.  To evaluate answer quality, we used GPT-4 to assess the generated responses, as shown in **Figure 12(a)** of the supplementary material or the **table below**. The results indicate that the answers produced by our method are of comparable quality to ground-truth answers. Notably, in most clusters, our model matches or exceeds the performance of the ground-truth answers, highlighting its effectiveness and reliability. At the same time, we acknowledge that human evaluation or user ratings would provide a more direct and reliable measure of how well the generated questions align with real user inquiries. While resource constraints have limited us in this regard, we recognize its importance and plan to incorporate human evaluation in future research.
>
> *[1] https://industrydocuments.ucsf.edu/wp-content/uploads/2022/05/MNK-TargetingHighVolumePrescribers.pdf*

---

> ### Author Response · Authors · 2024-11-25
> **GPT winning rate and persona ablation**
>
> **GPT winning rate results table**
>
> | Cluster | Our Answer Win | Tie   | Ground-Truth Win |
> |---------|----------------|-------|------------------|
> | 18      | 25.7           | 32.5  | 41.8             |
> | 19      | 17.9           | 26.1  | 56               |
> | 20      | 20.1           | 36.2  | 43.7             |
> | 29      | 16.9           | 32.5  | 50.6             |
> | 34      | 25.6           | 27.5  | 46.9             |
> | 38      | 25.4           | 31.5  | 43.1             |
> | 39      | 18.6           | 20.9  | 60.5             |
> | 65      | 22.2           | 29.4  | 48.4             |
> | 90      | 21.2           | 25.9  | 52.9             |
> | 94      | 21.1           | 27.9  | 51               |
> | **average** | 21.47          | 29.04 | 49.49            |
>
>
> **Persona ablation results table**
>
> | Cluster | Persona Win | Tie   | No Persona Win |
> |---------|-------------|-------|----------------|
> | 18      | 74.9        | 18.3  | 6.8            |
> | 19      | 73.1        | 19.5  | 7.4            |
> | 20      | 73.7        | 17.6  | 8.7            |
> | 29      | 72.1        | 19.7  | 8.2            |
> | 34      | 75.1        | 17.5  | 7.4            |
> | 38      | 75.6        | 17.5  | 6.9            |
> | 39      | 74          | 18.5  | 7.5            |
> | 65      | 72.1        | 19.6  | 8.3            |
> | 90      | 75.7        | 16.7  | 7.6            |
> | 94      | 75.7        | 16.1  | 8.2            |
> | **average** | 74.2        | 18.1  | 7.7            |

---

> ### Author Response · Authors · 2024-11-25
> **Author Response to Q2 and Q3**
>
> ## Q2: Effect of using these different modalities and ablation study
>
> We have conducted an ablation study using LoRA to evaluate the impact of different modalities—text, layout, page, and tags—on model performance. The results, presented in **Table 1 or table below**, show that the text-only model achieves the highest average performance on BERTScore. In contrast, incorporating layout information tends to reduce performance, likely because LLMs are primarily pre-trained on pure text, making it challenging to effectively learn and utilize layout features during fine-tuning. These findings highlight the dominant role of text in generation quality and suggest that integrating other modalities requires further exploration to balance their contribution to downstream model performance and user satisfaction.
>
> | Text | Layout | Tag | Page Information | Average BERTScore |
> |------|--------|-----|------------------|-----------|
> | ✔    | ✔      | ✔   | ✔               | 74.42     |
> | ✔    | ✔      | ✔   | -               | 74.86     |
> | ✔    | ✔      | -   | -               | 74.75     |
> | ✔    | -      | ✔   | -               | 75.69     |
> | ✔    | -      | -   | -               | 75.74     |
>
> While for some special clusters like cluster 11, which comprises various instructional and informational categories such as guides, tips, and tutorials, layout information proves beneficial due to the structured nature of these documents (as shown in **table below**).
>
> | Text | Layout | Tag | Page Information | Cluster 11 BERTScore |
> |------|--------|-----|------------------|-----------|
> | ✔    | ✔      | ✔   | ✔               | 77.69     |
> | ✔    | ✔      | ✔   | -               | 77.71     |
> | ✔    | ✔      | -   | -               | 77.71     |
> | ✔    | -      | ✔   | -               | 77.63     |
> | ✔    | -      | -   | -               | 77.57     |
>
> Similarly, as shown results of cluster 9 in **table below**, the inclusion of tag information extracted using the CLIP model improves performance, as this cluster features a significant number of visual elements.
>
> | Text | Layout | Tag | Page Information | Cluster 9 BERTScore |
> |------|--------|-----|------------------|-----------|
> | ✔    | ✔      | ✔   | ✔               | 74.20     |
> | ✔    | ✔      | ✔   | -               | 74.37     |
> | ✔    | ✔      | -   | -               | 74.39     |
> | ✔    | -      | ✔   | -               | 75.14     |
> | ✔    | -      | -   | -               | 75.04     |
>
>
> ## Q3: Generalizable and Limitations
>
> Our proposed framework is specifically designed for the OIDA, with the primary goal of leveraging an innovative AI assistant to extract actionable insights, identify misuse patterns, and support targeted interventions. This specialization makes it clear that the framework is not a general-purpose QA system but a domain-specific solution tailored to the unique challenges and characteristics of opioid-related data. As a result, its direct applicability to other domains is inherently limited.
>
> However, the methodologies presented in this paper—such as data collection strategies, multimodal integration techniques, and cluster-based model design—are highly generalizable and applicable to various domains. For example, the approach of integrating textual, visual, and layout information into a multimodal framework is versatile and can be broadly used in document understanding tasks. Additionally, while the benchmark dataset is designed to address the opioid crisis, its foundational methodologies were developed with general scanned PDF documents in mind. This broader context ensures that the models and insights derived from this research are relevant beyond this specific domain. By advancing these general methodologies while maintaining a specialized focus on the opioid crisis, the framework balances domain-specific impact with broader applicability, making it a valuable contribution to both focused and general multimodal research.

---

### Official Review · Reviewer_hVUW · 2024-11-05

**Soundness:** 2
**Presentation:** 3
**Contribution:** 2
**Rating:** 5
**Confidence:** 3

**Summary:**

This work aim at supporting research and the developments of tools to support health systems with opioid-related tasks. The authors introduce OIDA-QA, a new question-answering (QA) benchmark focused on opioid data, drawn from the Opioid Industry Documents Archive (OIDA). This multimodal QA benchmark includes data enrichment techniques to make the unstructured content of the OIDA's scanned and complex PDF documents more accessible. In addition, the benchmark incorporate persona-based question generation, simulating user questions from different professional and personal backgrounds. The model training process is optimized with LoRA to create a low-cost and scalable system.

**Strengths:**

New QA benchmark for a specific (largely unexplored) domain where there is no or little open-data available.

**Weaknesses:**

To build the datasets authors made choice that are not fully explained in the paper. More details would help the reader to get a full understanding of the building process (see my questions below).

**Questions:**

- Is there any previous work where persona hub was used? From the paper, it is unclear to me to understand the impact of using it and what type of errors can be generated.
- Evaluation is done using mostly overlapping metrics. While the field doesn't have yet a unique way of evaluating LLM outputs, I believe that some automatic metric would be helpful to assess other dimensions (faithfulness, correctness etc) beyond word overlap.
- Authors introduced the dataset as a resource to have models tailored to specific data types and detailed annotations. However, I found two things missing: 1) a more broaden description of possible real-world use cases that can benefit from this dataset, 2) an evaluation zero-shot using large SOTA models (do we know to which extend this information is already available in the parametric knowledge of large models?).

---

> ### Author Response · Authors · 2024-11-25
> **Author Response to Q1**
>
> We sincerely appreciate the reviewer’s thoughtful feedback and insightful suggestions.
>
> ## Q1: Persona hub
> Yes, we adopt the personas introduced in the work **[1]**, a collection of 1 billion diverse personas designed to encapsulate a wide range of perspectives. These personas act as distributed carriers of world knowledge, effectively tapping into the vast array of viewpoints embedded within LLMs. This capability enables the scalable generation of diverse synthetic data across various scenarios. In our work, we leverage these personas to enhance the diversity and quality of question generation. By incorporating personas, the generated questions reflect a broader range of perspectives, leading to improved question quality. The effectiveness of this approach is demonstrated in the ablation study provided in **Figure 12 (b)** of the supplementary material or **following table**, which compares question generation with and without personas. The results clearly highlight the value of incorporating personas in our methodology.
>
> | Cluster | Persona Win | Tie   | No Persona Win |
> |---------|-------------|-------|----------------|
> | 18      | 74.9        | 18.3  | 6.8            |
> | 19      | 73.1        | 19.5  | 7.4            |
> | 20      | 73.7        | 17.6  | 8.7            |
> | 29      | 72.1        | 19.7  | 8.2            |
> | 34      | 75.1        | 17.5  | 7.4            |
> | 38      | 75.6        | 17.5  | 6.9            |
> | 39      | 74          | 18.5  | 7.5            |
> | 65      | 72.1        | 19.6  | 8.3            |
> | 90      | 75.7        | 16.7  | 7.6            |
> | 94      | 75.7        | 16.1  | 8.2            |
> | average | 74.2        | 18.1  | 7.7            |
>
>
> Additionally, we provide some exact examples in **Figure 10** and **Figure 11** at supplementary. We show the question answering pairs with multiple personas with different professional background.
>
> *[1] Scaling Synthetic Data Creation with 1,000,000,000 Personas*

---

> ### Author Response · Authors · 2024-11-25
> **Author Response to Q2 and Q3**
>
> ## Q2: Metrics
>
> Thank you for your insightful suggestion. We appreciate the importance of evaluating dimensions beyond word overlap, such as faithfulness and factual correctness. In our work, we introduced sentence-level metrics to establish a general benchmark for evaluation. To further address your concern, we have also evaluated the faithfulness and factual correctness of the generated answers on the test dataset using RAGAS **[2]**, as shown in **Figure 13** of the supplementary materials or **following table**. The results demonstrate that our model performs well on both metrics. Additionally, the BertScore results presented in **Figure 5(d)** further verify the correctness of our model's outputs. These evaluations underscore the robustness of our approach in capturing meaningful and accurate answers.
>
> | Cluster | Faithfulness | Factual Correctness |
> |---------|--------------|---------------------|
> | 18      | 0.483        | 0.486              |
> | 19      | 0.340        | 0.359              |
> | 20      | 0.426        | 0.390              |
> | 29      | 0.423        | 0.436              |
> | 34      | 0.425        | 0.406              |
> | 38      | 0.474        | 0.438              |
> | 39      | 0.404        | 0.417              |
> | 65      | 0.430        | 0.468              |
> | 90      | 0.404        | 0.394              |
> | 94      | 0.445        | 0.392              |
> | **average** | 0.425        | 0.419              |
>
>
> *[2] Ragas: Objective metrics, intelligent test generation, and data-driven insights for LLM apps*
>
>
> ## Q3: Broaden Description & Zero-Shot
>
> In the Introduction, we have broadly discussed how our dataset contributes to general document understanding (e.g., OCR, detection, segmentation) and how it serves as a critical enabler for healthcare applications. Here, we would like to further highlight the real-world value of our dataset. Constructing a densely annotated QA dataset not only enhances general document understanding but also serves as a critical resource for healthcare applications. A key practical application lies in enabling an AI assistant system capable of efficiently processing the OIDA dataset to extract valuable insights. This system can help users quickly understand critical information about the opioid crisis, including patterns, trends, and actionable insights. By enabling rapid comprehension and data-driven decision-making, the AI assistant can become an indispensable tool for policymakers, healthcare professionals, and community leaders in addressing the opioid crisis more effectively. In the updated version, we will elaborate on additional real-world use cases that can benefit from this dataset.
>
> Regarding zero-shot evaluation, the results are presented in **Figure 5**, highlighted in deep blue (or zero-shot BERTScore results at **table below**). These results illustrate the performance of recent models without any fine-tuning on OIDA-QA. While these models demonstrate baseline capabilities, the results reveal significant gaps in domain-specific knowledge encapsulated in our dataset. After LoRA fine-tuning, substantial performance improvements are observed across clusters (**Figure 5, highlighted in pink** or LoRA BERTScore results at **table below**), underscoring the necessity of fine-tuning with our benchmark. This highlights the dataset’s value in equipping models with domain expertise, enabling tailored solutions for public health analysis, law enforcement, and crisis management, and addressing real-world challenges effectively.
>
> | Cluster | BERTScore (zero-shot) | BERTScore (LoRA) |
> |---------|-----------------------|------------------|
> | 18      | 70.7                 | 71.3            |
> | 19      | 67.6                 | 63.8            |
> | 20      | 67.4                 | 71.7            |
> | 29      | 69.9                 | 71.2            |
> | 34      | 68.6                 | 70.9            |
> | 38      | 68.9                 | 74.2            |
> | 39      | 66.9                 | 69.4            |
> | 65      | 68.8                 | 70.7            |
> | 90      | 68.7                 | 67.7            |
> | 94      | 67.8                 | 70              |
> | **Average** | 68.53                | **70.09**           |

---

> > ### Comment · Reviewer_hVUW · 2024-11-27
> >
> > Tank you for the effort in clarifying my concerns. After reading other reviews and your response, there is no doubt that the paper has its merits, but I'll keep my score because given the amount of changes required, this work would benefit from another round of reviews after all the suggestions are integrated into the paper.

---

> > > ### Author Response · Authors · 2024-11-28
> > > **Thanks**
> > >
> > > Thank you for taking the time to review our feedback and revised paper, and for recognizing the merits of our work. We greatly appreciate your thoughtful comments. In the updated manuscript, we have carefully incorporated your suggestions and addressed the issues raised, as we believe these changes enhance the overall quality and rigor of the study.
> > >
> > > We hope the feedback and additional results included in the revised manuscript demonstrate our commitment to improving this work and advancing the understanding of the opioid crisis. Thank you again for your constructive feedback and valuable contributions to this discussion.

---

### Official Review · Reviewer_9CD6 · 2024-11-08

**Soundness:** 2
**Presentation:** 3
**Contribution:** 3
**Rating:** 5
**Confidence:** 3

**Summary:**

This paper introduces OIDA-QA, a new multimodal question-answering benchmark designed to analyze the Opioid Industry Documents Archive (OIDA). Acknowledging the complexity and multimodal nature of the data—including text, images, and layout—the authors reorganize the original dataset into training documents and testing documents. They extract comprehensive multimodal information from each document to capture diverse features. By generating over 3 million question-answer pairs using AI models and incorporating persona-based question generation, they aim to simulate diverse user interactions. The authors develop domain-specific LLMs using LoRA, focusing on cluster-specific adaptation to enhance performance. Experimental results demonstrate improvements in document information extraction and question-answering tasks, suggesting that their approach effectively addresses challenges associated with the opioid crisis.

**Strengths:**

1. The paper presents a novel benchmark tailored to the opioid crisis, addressing a timely public health issue. By focusing on the OIDA, the authors provide a unique resource for analyzing complex, multimodal healthcare data.

2. The methodology is thorough and well-executed. The authors analyze the dataset's distribution, perform balanced sampling, and extract detailed multimodal information.

**Weaknesses:**

1. The experimental results primarily focus on quantitative metrics, leaving out an in-depth qualitative evaluation of the model outputs. Including case studies or practical examples demonstrating the effectiveness and real-world applicability of the generated answers would enhance the paper's impact and provide a clearer understanding of the model's performance.

2. The paper does not sufficiently explain the rationale behind simulating users through diverse personas encompassing various backgrounds, experiences, and hobbies. The inclusion of personas such as Music Producer and Professional Composer (in the Appendix) raises questions about their relevance to the task. A more thorough justification is needed to clarify how these personas contribute to the dataset and the overall objectives.

3. The data generation process relies entirely on GPT-4 models. To ensure the validity, reliability, and quality of the dataset—particularly the test set—it is important to include human (expert) evaluation.

4. Apart from using documents from the Opioid Industry Documents Archive (OIDA), the paper does not introduce any novel methodologies specifically designed to address the opioid crisis. The approach seems to be a general application of multimodal document question-answering techniques, which could equally be applied to other datasets such as PubMed articles. This raises concerns about the paper's unique contribution to tackling the opioid crisis beyond dataset curation.

**Questions:**

Please refer to the Weaknesses.

---

> ### Author Response · Authors · 2024-11-25
> **Author Response to Q1 and Q2**
>
> We thank the reviewer for reviewing our paper and providing valuable suggestions.
>
> ## Q1: Evaluation and Real-World Applicability
>
> Thank you for your insightful feedback. This paper aims to provide a foundational dataset for large-scale PDF parsing and QA tasks, along with a robust quantitative benchmark for future comparisons. While we acknowledge that real-world case studies would enhance the practical relevance of our approach, our focus in this work is on addressing the core challenges of parsing massive PDF data and building the benchmarking framework. In particular, we recognize that healthcare applications are highly specialized, requiring rigorous data collection and strict ethical compliance. Nevertheless, we appreciate your suggestion and have included visualized examples in the supplementary material (**Fig. 10 and 11** in the Appendix) to illustrate how our method could potentially address real-world applications.
>
> As demonstrated by the models in our paper, our dataset also lays the groundwork for developing AI systems capable of analyzing the OIDA dataset to extract actionable insights, such as identifying opioid misuse patterns, tracking overdose trends, and highlighting regional disparities. These systems could assist healthcare professionals in identifying high-risk areas and support policymakers in implementing data-driven interventions. To further facilitate advancements in the healthcare domain, we will release the dataset and models to encourage broader adoption and improvement.
>
>
> ## Q2: Persona Justification
>
> Thank you for your insightful question. The design of personas was informed by discussions with professionals in the medical field, with the goal of creating a dataset that is broadly applicable and beneficial to a diverse audience. A strictly user-specific QA approach for this data could limit the model’s generalizability and reduce its accessibility for a wider user base.
>
> To further clarify the rationale behind simulating users through diverse personas, we provide additional results in **Figure 12(b)** of the updated supplementary or **following table**. These results are based on an ablation study examining the role of personas in question generation. Evaluations conducted using GPT-4 demonstrate that incorporating personas significantly enhances question quality, with questions generated using personas consistently outperforming those generated without them. This underscores the critical role of personas in improving question generation and aligning with the dataset’s broader objectives.
>
> | Cluster | Persona Win | Tie   | No Persona Win |
> |---------|-------------|-------|----------------|
> | 18      | 74.9        | 18.3  | 6.8            |
> | 19      | 73.1        | 19.5  | 7.4            |
> | 20      | 73.7        | 17.6  | 8.7            |
> | 29      | 72.1        | 19.7  | 8.2            |
> | 34      | 75.1        | 17.5  | 7.4            |
> | 38      | 75.6        | 17.5  | 6.9            |
> | 39      | 74          | 18.5  | 7.5            |
> | 65      | 72.1        | 19.6  | 8.3            |
> | 90      | 75.7        | 16.7  | 7.6            |
> | 94      | 75.7        | 16.1  | 8.2            |
> | **average** | 74.2        | 18.1  | 7.7            |

---

> ### Author Response · Authors · 2024-11-25
> **Response to Q3 and Q4**
>
> ## Q3: Expert Evaluation
> Thank you for your valuable suggestion. We agree that incorporating human (expert) evaluation would enhance the validity, reliability, and overall quality of the dataset, particularly for the test set. However, the primary challenges for most PDF-based document datasets are the data source and metadata limitations. These challenges motivated us to create a densely annotated resource encompassing OCR, detection, segmentation, and a QA benchmark, complemented by baseline models.
>
> While expert human evaluation is undoubtedly valuable, due to time constraints, we opted to utilize GPT-4 to evaluate the answers generated by our model. As shown in **Figure 12(a)** of the supplementary material or **following table**, the evaluation results indicate that our model achieves answer quality comparable to ground-truth responses. In fact, across many clusters, our model performs on par with or even surpasses ground-truth answers, highlighting its effectiveness and reliability. We plan to include human evaluation experiments in our future work.
>
> | Cluster | Our Answer Win | Tie   | Ground-Truth Win |
> |---------|----------------|-------|------------------|
> | 18      | 25.7           | 32.5  | 41.8             |
> | 19      | 17.9           | 26.1  | 56               |
> | 20      | 20.1           | 36.2  | 43.7             |
> | 29      | 16.9           | 32.5  | 50.6             |
> | 34      | 25.6           | 27.5  | 46.9             |
> | 38      | 25.4           | 31.5  | 43.1             |
> | 39      | 18.6           | 20.9  | 60.5             |
> | 65      | 22.2           | 29.4  | 48.4             |
> | 90      | 21.2           | 25.9  | 52.9             |
> | 94      | 21.1           | 27.9  | 51               |
> | **average** | 21.47          | 29.04 | 49.49            |
>
>
> ## Q4: Methodologies Designed to Address the Opioid Crisis
>
> We would like to highlight the unique contributions of our work, particularly in comparison to PubMed datasets.
>
> *Granular and Multimodal Data Construction*： Compared to PubMed, our work introduces several distinctive features:
> - Granular Data Construction: Data is extracted at four levels of granularity—text line, paragraph, doc2box, and doc2mask (as shown in **Figure 2**)—providing a detailed and flexible framework for diverse research tasks.
> - Multi-Page and Multimodal Context: Our dataset is designed to handle multi-page documents with multimodal content, addressing challenges in complex document analysis.
> - Persona-Based QA Generation: By incorporating persona diversity, our dataset enhances the contextual depth and variety of question-answer pairs, fostering richer research insights.
>
> *Focused Application to the Opioid Crisis*： While our techniques are broadly applicable, the dataset and LLM-based AI assistant are specifically designed to address the opioid crisis. By leveraging domain-specific data (OIDA), the assistant identifies opioid misuse patterns, informs targeted interventions, and supports public health and policy efforts. This moves beyond general dataset curation to provide actionable tools for combating this critical issue.

---

### Meta-Review · Area_Chair_8ahK · 2024-12-21

**Metareview:**

The paper introduces OIDA-QA, a multimodal benchmark leveraging AI models to analyze opioid industry documents, aiming to enhance question-answering and public health insights on the opioid crisis.

Strengths
- The reviewers appreciate the introduction of a new multimodal QA benchmark tailored to the opioid crisis, addressing an important public health issue
- Reviewers appreciate the use of personas to generate diverse QA pairs

Weaknesses
- The paper relies entirely on AI-based evaluations, with no human expert validation to ensure dataset and QA quality
- Limited qualitative analysis and lack of real-world case studies limits the paper's practical impact
- The applied methods are generalizable to other datasets, raising questions about their uniqueness for addressing the opioid crisis specifically

**Additional Comments On Reviewer Discussion:**

During the rebuttal, the authors failed to resolve the reviewer concern regarding the lack of human expert validation. The authors promised this in the future work.

---

### Decision · Program_Chairs · 2025-01-22

Reject